# Softness or Stiffness What Contributes to Cancer and Cancer Metastasis?

**DOI:** 10.3390/cells14080584

**Published:** 2025-04-12

**Authors:** Claudia Tanja Mierke

**Affiliations:** Faculty of Physics and Earth System Sciences, Peter Debye Institute of Soft Matter Physics, Biological Physics Division, Leipzig University, 04103 Leipzig, Germany; claudia.mierke@uni-leipzig.de

**Keywords:** cell mechanics, physics of cancer, viscoelasticity, softness and stiffness, matrix confinement, mechanobiology, deformability, optical cell stretcher, atomic force microscopy, tissue patchiness

## Abstract

Beyond the genomic and proteomic analysis of bulk and single cancer cells, a new focus of cancer research is emerging that is based on the mechanical analysis of cancer cells. Therefore, several biophysical techniques have been developed and adapted. The characterization of cancer cells, like human cancer cell lines, started with their mechanical characterization at mostly a single timepoint. A universal hypothesis has been proposed that cancer cells need to be softer to migrate and invade tissues and subsequently metastasize in targeted organs. Thus, the softness of cancer cells has been suggested to serve as a universal physical marker for the malignancy of cancer types. However, it has turned out that there exists the opposite phenomenon, namely that stiffer cancer cells are more migratory and invasive and therefore lead to more metastases. These contradictory results question the universality of the role of softness of cancer cells in the malignant progression of cancers. Another problem is that the various biophysical techniques used can affect the mechanical properties of cancer cells, making it even more difficult to compare the results of different studies. Apart from the instrumentation, the culture and measurement conditions of the cancer cells can influence the mechanical measurements. The review highlights the main advances of the mechanical characterization of cancer cells, discusses the strength and weaknesses of the approaches, and questions whether the passive mechanical characterization of cancer cells is still state-of-the art. Besides the cell models, conditions and biophysical setups, the role of the microenvironment on the mechanical characteristics of cancer cells is presented and debated. Finally, combinatorial approaches to determine the malignant potential of tumors, such as the involvement of the ECM, the cells in a homogeneous or heterogeneous association, or biological multi-omics analyses, together with the dynamic-mechanical analysis of cancer cells, are highlighted as new frontiers of research.

## 1. Introduction

Cancer is still a deadly disease, where at least over 60% of cancer deaths are due to metastatic spread of the primary tumor [1]. Sequencing of the human genome and advancements in genetic and molecular analyses have not led to major breakthroughs in cancer that could significantly reduce metastasis-related cancer deaths [2]. Therefore, there is still a need to gain an in-depth understanding of cancer development, especially the malignant progression of cancer such as metastatic spread, in the effort to reduce mortality due to metastasis. The mechanical analysis of cells has attracted a lot of attention and has found its way into cancer research. Consequently, an interdisciplinary field of the physics of cancer has emerged. A hotly debated topic is whether the mechanical properties of cancer cells are a prerequisite for the malignant progression of cancers or whether they are merely a consequence of it. Regardless of the answer to this question, the mechanical properties of cancer cells have been investigated using various biophysical methods, some of which have been newly developed [3] or adapted specifically for the analysis of living cells [4,5]. Nevertheless, the mechanical characteristics of cancer and cancer cells seem to be important in understanding the complex nature of cancer development and malignant progression such as cancer metastasis. Research into the mechanical nature of cancers indicates several aspects that could play a role. Firstly, when characterizing the mechanical properties of cancer cells, such as stiffness/softness, the question arises as to whether the focus is on cancer cells, solid tumors, or the tumor microenvironment (TME). Secondly, the biophysical technique for determining stiffness/softness is important because several techniques work with adhesive and non-adhesive cells, probe cells broadly, or at specific sites via specific receptors, enable repeated mechanical probing and high or low throughput, include environmental cues such as a 3D environment, or stimulate cells before (cell culture and preparation conditions) and/or during the measurement (forces on cells, heat, flow conditions or marker beads). Thirdly, the mechanical testing of the mechanical properties of the entire cell mass or subcellular units. Fourthly, what types of cancer are investigated, and which model system is selected (e.g., samples of primary cancer cells or cancer cell lines)? Fifthly, the cultivation conditions before and during the mechanical characterization could have an influence on the mechanical properties of the cancer cells. Finally, the hotly debated question of whether the mechanical properties of cancer cells are a prerequisite or a consequence of the malignant progression of cancers must be answered.

One emphasis of research on the physics of cancer has been placed on the characterization of cancer cells with various biophysical techniques such as atomic force microscopy (AFM), optical cell stretchers, optical tweezers, magnetic tweezers, nano-scale particle tracking, traction force microscopy, and microfluidic constriction channels. Apart from the different approaches of measuring the mechanical characteristics of cancer cells, the question must be raised as to whether it is at all advantageous for cancer cells to exhibit specific mechanical characteristics. In particular, the following question needs to be answered: Why is it an advantage of cancer cells to become softer compared with normal healthy cells? These aspects will be debated in this review. In addition, whether the mere analysis of the passive mechanical properties of cancer cells is sufficient to serve as a mechanical marker for cancer cells, especially malignant cancer cells, will be discussed. Alternative approaches and future directions will also be discussed. In addition, the role of the microenvironment is highlighted.

## 2. Mechanical Properties of Cancer Cells with a Focus on Softness/Stiffness

Naturally, cells have certain properties, one of which is elasticity. Elasticity refers to the capacity of a material to resume its original shape when an external force is discontinued. Elasticity is also defined as the capacity of a material to resist deformation under stress [6,7]. Conversely, stiffness is a measure of the elasticity of a material and is given by the stress (force per unit area) required to cause a specified strain (deformation) [6,8,9]. The cell itself is an extremely heterogeneous material, and it is advantageous to assess the mechanical characteristics as a whole rather than to measure the cell’s local elasticity [10]. In the following, the generality of the softness of metastatic and aggressive cancer cells is discussed. Moreover, the difficulties in the detection of soft cancer cells are presented. Thereafter, the impact of the 3D microenvironment on the mechanical properties of cancer cells is discussed.

### 2.1. Classical Approach for Mechanical Characterization of Cancer Cells

The classical approach for analyzing the mechanical properties of cancer focuses on cellular mechanical cues such as the deformability or softness of cells (inverse stiffness). It has been seen that more deformable cells adhere stronger to other normal cells (Table 1) [11]. Most biophysical measurements are typically performed as static analyses at a single point in time, and the initial mechanical characterization of cancer cells has been carried out using AFM [4].

As measurements of individual living cancer cells using AFM are time-consuming and hence rather low throughput, other biophysical techniques of elevated throughput have been utilized. The measurement of individual cells in a suspended state, such as in biophysical techniques based on microfluidic chambers like optical cell stretcher analysis, seems deceptively straightforward [30]. These experiments are carried out without considering the differences in the cell adhesion strength within a cell population, between different cancer cell types, or between cancer cells and healthy control cells. As the variations between individual cancer cells are large, a high number of cells is usually analyzed. It is not known how many cancer cells in a primary tumor population must change their mechanical properties for malignant progression of the tumor disease and metastasis to occur. It is possible that the cells that lead to malignant progression and metastasis are overlooked in the mass of cells analyzed, especially when the number of measured cells is averaged. A question has therefore arisen as to how useful such measurements are. Regardless of whether these measurements are useful or not, care has been and is still taken to ensure that as many cells as possible are analyzed at a high throughput. The traditional approach of achieving a high throughput of individually analyzed cells has therefore been pursued, which can usually be achieved using microfluidic, flow-based approaches such as optical cell stretching [3,31,32]. The decline in cancer cell stiffness has been connected to certain cancerous traits such as unrestrained proliferation, the avoidance of apoptosis, and heightened motility [19,33,34,35,36,37]. The hypothesis that softer cancer cells are more malignant is intuitive. The capacity of cancer cells to infiltrate tissues and form metastatic foci at distant locations depends on their ability to surmount mechanical barriers that arise from traveling through narrow gaps within the extracellular matrix (ECM) and intercellular spaces [38,39,40]. The growing realization that cancer cells may have to adjust their mechanics to advance disease has led to significant efforts to elucidate modified cellular mechanics as physical biomarkers and mechanotransductive mechanisms as therapeutic drug targets. A large number of these investigations have concentrated on detecting variations in the mechanical characteristics of cancer cells in comparison with their corresponding non-tumorigenic cells [4,17,27,28,41,42,43,44,45,46,47]. In addition to the capability of cancer cells to deform themselves to pass through tissue bottlenecks, the softness of cancer cells confers an advantage on these cells with respect to the immune response of the organism.

### 2.2. Impact of Softness/Stiffness of Cancer Cells on Immune Response

Cancer cells and solid cancers can shape the immune response through their cellular mechanical cues [48,49,50]. The progression of cancer is associated with extensive alterations in cellular mechanics. This finding implies that immunosurveillance has acquired a mechanical component, which is referred to as mechanosurveillance [51,52,53]. Cancer cells can even cause a shift in the polarization of immune cells, such as macrophages or neutrophiles or stromal cells like cancer-associated fibroblasts (CAFs), whereby the stromal cells then remodel the TME by accumulation, crosslinking, and contraction, which subsequently typically becomes stiffer (Figure 1) [54,55].

Consequently, a solid castle-like barrier of mechanical stiffness is formed that shields the solid primary tumor from the environmental, biological, and mechanical signals [56,57]. Moreover, stiffer solid tumors exhibit less immune cell abundance within their core region [58,59]. In addition, the activity of the immune cells is impacted by the cancer cell stiffness [60]. For instance, the softness of cancer cells acts as a mechanical immune checkpoint and attenuates tumor-specific T-cell-driven immunotherapy [61]. In general, it has been proposed that malignant cancer cells exhibit altered mechanical properties in comparison to their non-malignant equivalents. A manifestation of these altered mechanical characteristics in malignant cells can be seen in their markedly lower stiffness (softness) compared with their benign equivalents, a finding that holds true for a variety of cancers such as breast cancer, bladder cancer, cervical cancer, colorectal cancer and thyroid cancer [3,4,12,17,18,20,24,29,62]. The softness of cancer is strongly related to tumor transformation and aggressiveness [63]. This close connection arises from the lymphocyte-facilitated immune attack, which relies heavily on direct immune synapse interactions with target cells such as cancer cells (Figure 2) [64]. When activated during the attack, these lymphocytes release toxic perforins or granzymes into their synaptic space, thereby causing apoptosis of the target cancer cells [65]. The softening of cancer cells usually affects the activation of lymphocytes, leading to decreased levels of potent effector cytokines secreted by lymphocytes [66]. This means that the softness of cancer cells acts as a mechanical immune checkpoint to trigger an immune resistance to lymphocyte-driven cytotoxicity [67]. Cholesterol in the plasma membrane has turned out to be important for regulating cell stiffness [68]. Cancer cells usually have elevated cholesterol levels in their plasma membrane compared with normal cells, which causes cancer cells to become softer [61]. An increase in the stiffness of cancer cells can render them more susceptible for attack by lymphocytes (Figure 2) [69]. Cancer cell membranes usually have higher cholesterol levels compared with normal cells, which renders cancer cells soft. Subsequently, a mechanical immune checkpoint inhibitor (MICI) has been developed, which consists of cyclodextrin lipids and fusogenic lipids, with which it is possible to provide functional evidence for this connection [69].

By fusing cyclodextrin lipids into the cancer cell membrane by utilizing a fusogenic liposome delivery system, the cholesterol in the plasma membrane of cancer cells is decreased as a result of the specific host–guest interplay between the cyclodextrin lipid and cholesterol [69]. Thereby, cancer cells are stiffened, and the activation of lymphocytes involving NK and cytotoxic effector T cells is enhanced when they encounter the stiffened cancer cells, as evidenced by strong degranulation and effector cytokine generation. In particular, cytotoxic synapses exhibit mechanical activity, exerting forces in the nanonewton regime on the target cell such as cancer cells [70]. These forces are involved in both the activation of mechanosensitive cell surface receptors on the lymphocytes and the consequent enhancement in perforin activity [71,72,73,74,75,76,77]. Exocytosis of lytic granules, referred to as degranulation, generally occurs at sites of active force exertion inside the immune synapse [71], suggesting that local mechanosensation may instructively guide perforin and granzyme liberation. This hypothesis is especially interesting because a number of receptors with a proven function in the generation of immune synapses, such as the T-cell receptor (TCR), several activating NK receptors, and the integrin lymphocyte function-associated antigen 1 (LFA-1), can be considered mechanosensitive [77,78,79,80]. Remarkably, this approach has minimal impact on lymphocyte invasion and proliferation within tumor tissue, supporting the notion that the increased anti-tumor efficacy is due to the activation of a subset of lymphocytes through the direct control of cancer cell stiffness. Like other integrins, LFA-1 exists in several conformational states [81], which differ in their affinity for the ligands, namely the intracellular adhesion molecules (ICAMs) 1 and 2. In its resting bent conformation, LFA-1 has low ligand-binding activity. “Inside-out” activation signals, which are usually triggered by activating receptors such as TCR, drive the elongation of LFA-1 toward an intermediate state of affinity [82,83]. LFA-1 must be linked to the cortical filamentous actin (F-actin) cytoskeleton for optimal ligand engagement [84,85,86]. This coupling tensions the integrin, facilitating both high-affinity catch-bonding with ICAM and “outside-in” signaling complexes assembled at its tail domains [87]. In this way, LFA-1 strongly adheres to the target cell and simultaneously produces co-stimulatory signals that enhance the activation of lymphocytes [88,89,90,91]. In conclusion, it can be proposed that the softness of cancer cells may help them avoid the activation of immune cells. Therefore, analyzing the mechanical softness/stiffness of cancer cells appears useful, and a hypothesis has been put forward that softer cancer cells may lead to a general worsening of the disease and to metastases across all tumor types.

### 2.3. Are Softer Cancer Cells Generally More Malignant?

Since specific types of cancer cells have been revealed to be softer compared with their normal healthy equivalents (see Table 1), it has been proposed that cancer cells may become softer when they invade the ECM scaffold and undergo metastasis [34]. This means that cancer cells soften when migrating and invading through a spatial confinement. Although intuitively compelling, the so-called adaptive softening hypothesis relies primarily on comparisons of normal and cancer cells at the population level [3,20,30,92,93,94]. AFM measurements, for instance, have shown that human bladder cancer cells are an order of magnitude softer compared with healthy bladder cells [4]. Optical stretching measurements at high throughput also suggest that human breast cancer cells undergo more severe deformation compared with normal breast epithelial cells, and that the deformability rises with the metastatic capacity [3]. Therefore, another hypothesis was proposed that states that there exists a certain subgroup of cancer cells within the cancer cell population that become even softer compared with their neighboring cancer cells [23,63]. The picture that emerges from these population-based studies is not entirely clear-cut. For instance, AFM measurements have demonstrated that leukemic cells are stiffer compared with leukocytes [95], and micropipette aspiration experiments revealed that hepatocellular carcinoma cells are stiffer compared with normal hepatocytes. [41]. A large body of other work using a range of mechanical approaches supports evidence that cancer cells are either stiffer compared with their normal counterparts or stiffen during tumor development, challenging the adaptive softening hypothesis [96,97,98]. In line with this, chondrosarcoma cells have been found to be stiffer when compared with chondrocytes, whereby chondrosarcoma cells display markedly elevated β-tubulin expression (Table 2) [26].

### 2.4. Difficulties in Detecting Soft Cancer Cells in Cancers and Impact of Environmental Cues on Cancer Cell Mechanics

There is no standard method to identify individual softer cancer cells within a solid cancer type. The identification of softer cancer cells may be hampered by the identification of apoptotic cancer cells that also become softer [107], but they do not contribute to cancer advancement via cancer metastasis. Therefore, how can we identify these early apoptotic cancer cells that do not express biological markers of apoptosis yet? The identification of morphologically altered cellular nuclei of cancer cells that will become highly aggressive does not provide an alternative path to overcome this obstacle, as apoptotic cells are more condensed and thus smaller in their nuclear size. Thus, a question has arisen as to how the nuclei of apoptotic cells can be distinguished from those of aggressive cancer cells. The nuclei of these aggressive cancer cells have been reported to be softer at a certain stage of aggressiveness, which has not yet been clearly determined. As the nuclei of apoptotic cells adapt their plasticity, they may therefore also become softer [108]. Consequently, there seems to be a further problem in distinguishing the nuclei of apoptotic cells from aggressive cancer cells.

Cancer cells may adapt to their mechanical environment and hence become stiffer due to elevated stiffness of their microenvironment, which causes an elevation in malfunctional mitochondria. These damaged mitochondria aggregate, and thus the entire cancer cell stiffens. Stiffening of cervical, hepatocellular cancer, and leukemia has been revealed [41,109,110]. In the case of defective autophagy processes, the injured aggregated mitochondria cannot be broken down and accumulate within the cancer cells, which then leads to elevated stiffness of these cancer cells. When these cancer cells manage to clear the damaged accumulated mitochondria, the cancer cells will become softer again. These softer cancer cells can withstand treatment via pharmacological drugs and acquire an advanced cancer cell state in which these cells become motile and invasive. The relationship between deformation and invasion has been reported for prostate cancer [28,111]. Another alternative mechanism could occur by which cancer cells are able to reacquire mitochondria from healthy immune cells [112]. How this contributes to their malignant behavior is still elusive. The transfer of mitochondria is unidirectional, whereby T cells hand over their mitochondria to cancer cells [112]. This is thought to improve the metabolism of cancer cells and deliver extra metabolic power to them. At the same time, the immune cells are depleted from mitochondria and lose metabolic energy, which seem to contribute to the immune escape of cancers, as has been described in the hallmarks of cancer. Moreover, the transfer of mitochondria to cancer cells is linked to enhanced activity of the cancer cell’s cell cycle, and therefore worse clinical prognosis in various types of cancer [112]. Apart from cancer cells, the TME stiffness can influence the activity of stromal cells, such as CAFs, which secrete tumor necrosis factor-β (TGF-β) [113].

## 3. Impact of Traditional Biophysical Techniques on Cancer Cell Mechanics

Some biophysical methods measure the mechanical properties of cells in a rather unphysiological way, since there is no cell adhesion and no microenvironment during the measurement process, even if the cells are adherent cells in their natural environment. The mechanical investigation of biological materials in the context of disease has received widespread recognition in the last few decades. Consequently, numerous techniques for investigating cell and tissue mechanics have been customized and developed. These biophysical techniques comprise, among others, AFM [114], optical cell stretchers [30,31], optical tweezers [115], micropipette aspiration (MPA) [116], microfluidic devices, and optical magnetic twisting cytometry (OMTC) (Figure 3).

### 3.1. AFM

Specifically, AFM has been very popular in this mechanobiological research area because it can be used in an aqueous surrounding and at a temperature that best mimics the natural environment of most biological probes, a feature not offered by other techniques for characterization [117,118]. The AFM equipment is composed of a cantilever that is pressed on the cells with a specific force (Figure 3). AFM has advantages in its capability to deliver high-resolution nanotopographic images and relatively uncomplicated sample processing, even though it requires special considerations for precise mechanical analysis [117,118,119,120]. In comparison to the majority of other biophysical techniques, however, AFM is characterized by low throughput, high time requirements, and technical difficulties [118,119,121]. Nevertheless, AFM has resulted in groundbreaking work that has expanded the scientific community’s understanding of cell and tissue mechanics, especially in the field of cancer. Most AFM research is focused on the nanomechanical investigation of cells, often in the context of cancer research [118,119]. The AFM technique can be used to distinguish between healthy cells and cancerous cells on the basis of their different mechanical characteristics [111]. In addition, this technique can be employed to differentiate between various cancer cell phases during cancer development and its malignant progression. Thus, the impact of cancer cell plasticity can be determined for each cancer type. For instance, when combined with machine learning and image processing approaches, AFM can distinguish two human colon cancer cell lines from each other based on the stiffness levels that are negatively correlated with neoplastic aggressiveness, whereby the more aggressive cell line is softer [122]. Nanotopographic analysis and mechanical characterization using AFM can be employed to elucidate the type of regulated cell death, for instance, intrinsic apoptosis, extrinsic apoptosis, necrosis, and ferroptosis, within a population of mouse fibrosarcoma cells [123].

Due to the outstanding sensitivity of microcantilevers, AFM-based microindentation is quickly becoming the favorite technique for investigating living cells. The limitations of AFM, such as the working distance, cell area analyzed, cantilever coating, and amount of force application, however, restrict its use in several research applications. These variations are mirrored in the reported values of the cellular modulus of elasticity, which vary by orders of magnitude. AFM is based on specific assumptions about the cell, which is soft, thin, irregularly formed, and immersed in liquid, complicating the interpretation of contact point and force–distance curves due to noise and the apparent absence of sharp discontinuities. Painstaking experiments that take these factors into account when measuring viscoelasticity are laborious, and the total number of cells that can be confidently examined may be restricted. Due to the low throughput of AFM measurements, the possibility of synchronizing measurements over a population or with different biological and biochemical phenotypic states is inevitably hampered (Table 3).

### 3.2. Optical Cell Stretcher

The optical cell stretcher was developed as an alternative technique for analyzing the mechanical properties of individual cancer cells [30]. For more details, please see the review article [32]. The throughput of this technique is elevated compared with AFM analysis, and the entire measurement of the cells can be performed in a more automated way than AFM. In particular, the optical cell stretcher is an optofluidic instrument that connects the microfluidic channel and the laser beams used for optical stretching (dual-beam laser trap). The two laser beams must be aligned perpendicular to the flow direction, where they need to be placed symmetrically to the channel centerline (Figure 3). The laser trap has been designed and constructed so that two identical counter-propagating beams intersect the microchannel, usually in the “lower hemisphere” of the channel, for instance at a height of 25 µm above the bottom, in order to gently trap the cells that are streaming into the channel [31]. Through laser radiation, a cell contact-free force is exerted on the cell’s surface (Table 3), which results in a deformation of the cells due to momentum transfer caused by light scattering at the front and back surfaces of the cells [3,30]. The amount of deformation of the cells relies on the mechanical characteristics of the cell probe [3]. The integrated microfluidic system enables a high level of efficiency when capturing and analyzing cells that flow through the channel. The cells are analyzed as non-adherent cells in a suspended state. Various investigations have indicated that the optical deformation of cells detected by an optical stretcher can be exploited as a mechanical signature to discriminate healthy cells from cancerous and metastatic ones as well as monitor the impact of pharmaceutical interventions on the mechanical reaction of the cell [3,30,132,133]. A weakness is that cells with dark granules cannot be measured, as the cells would explode within the microfluidic chamber.

### 3.3. Micropipette Aspiration (MPA)

In MPA, the biophysical measurement method is based on a negative pressure inside the micropipette that is applied to the (cancer) cell to produce a phenomenon known as an aspiration site on the cell (Figure 3). The local deformation of the plasma membrane of the cell in the contact area is then examined [93]. In cases where the exerted negative pressure on the cell is well-determined, the obtained optical images of the cellular deformation are exploited to quantitatively assess the distance the cell is sucked into the pipette, which is referred to as the aspiration length [134]. It is possible to determine the Young’s modulus from MPA measurements when a simple continuum model is utilized that assumes the cell to be a homogeneous, linear-elastic, and incompressible half-space [134,135]. With the MPA, the pressure distribution that can be exerted on the cells spans a wide area, starting at low forces of pN/μm^2^ and culminating in high forces of nN/μm^2^. These forces can be generated over several tens of µm^2^, enabling overall forces of up to some tens of nN to be exerted on the cells. Since many extensive alterations of the cell morphology are associated with a global force of several nN, MPA is a fascinating technique for investigating these phenomena [128,136,137]. In past decades, the MPA technique was frequently deployed to mechanically perturb cells and characterize the cell mechanics [138,139,140,141]. Using simple mechanics, the cortical tension and the elastic Young’s modulus of the sucked cell can be quantified [139]. As the MPA technique is relatively simple and easy to use, it can be combined with other methods such as genetics and fluorescent protein-labeling techniques. Combining MPA with cell biological techniques not only makes it possible to determine the mechanical properties of cells, but also to track locally what exactly takes place in the cells when the measured cell is mechanically stimulated at the site of measurement [142]. This technique also facilitates the application of mechanical stress to cells and the study of the cellular response to this mechanical disturbance (mechanosensing) by altering how proteins are distributed. Proteins like myosin II, which is a force-generating mechanoenzyme), and cortexillin I, which is an actin cross-linking protein in Dictyostelium discoideum, are concentrated in areas of high mechanical stress caused by MPA. [142,143]. Consequently, MPA provides a highly efficient technique that can be implemented to investigate cellular mechanics and mechanosensitive responses as well as to elucidate mechanoresponsive cellular signaling processes [142]. Finally, although the MPA technique is simple and classic, it is still useful for combined approaches with other techniques and thus has its merits (Table 3). Even though MPAs are relatively inexpensive to implement, they are naturally constrained due to their low spatial and temporal resolutions and the consequent large deformations acting on the cell [127]. Moreover, the quality of the cell-to-micropipette seal is important for achieving adequate and consistent data, but there is no general guideline for the acceptable quality of the seal, thus the choice is up to the individual user [144].

### 3.4. Microfluidic Devices

Microfluidic technologies can be used to analyze the mechanical deformation of cells when squeezing through a confinement as well as the electrical characteristics of individual non-adherent cells (Figure 3) [145]. The former technique is referred to as microfluidic cell transit analysis. Due to the microchannel geometry, it is feasible to monitor cells migrating through confined channels under different experimental conditions, or to visualize cellular deformations at various streaming velocities [146,147,148]. The aim is to make cells move through microfluidic contraction in long and narrow channels and to determine the mechanical characteristics by evaluating the deformation length and transit time for each individual cell within a population. Despite the possibility of collecting large amounts of experimental data, the measurement parameters are affected due to the adhesive characteristics of the cells, and the multi-parameter analysis cannot be verified qualitatively by using other experimental biophysical techniques [130]. The measured cells are suspended in a liquid and transported by it during the measurement process. It is important that the cells do not come into contact with each other or with the channel walls. In this sense, cells should not adhere to the channels or contaminate the channel surfaces, which can cause blockages. A microfluidic approach also allows for the magnitude of the hydrodynamic stresses to be adjusted. Therefore, it is possible to analyze sensitive cells with a spectrum of different degrees of deformability [147]. Thereby, various (cancer) cell types can be compared in a reliable manner. After penetration into the area of the extensional flow, a digital high-speed camera mounted on an inverted microscope captures multiple thousands of deformations in a second. The cells are then traced and tracked with an automated cell-image analysis algorithm to quantitate the initial cell diameter and cell deformability. The deformability of the cell is the long axis length of a deformed cell in relation to a perpendicular shorter axis length that is displayed in 2D flow cytometric scatter plots, as it is known from classical flow cytometric analyses [147]. This automation reduces user subjectivity and improves reproducibility (Table 3), which is a major concern when manual mechanical measurements are performed. The latter microfluidic technique involves electrical characterization, referred to as the µ-electrical impedance spectroscopy (µ-EIS) technique, which provides a viable means to study the dielectric properties of individual cells through the application of a frequency-dependent stimulating signal upon the cell. Moreover, the microfluidic confinement studies can be coupled with impedance measurements [130,149]. In summary, microfluidic deformability cytometry extends the statistical rigor of conventional flow cytometric procedures to unlabeled biophysical biomarkers, opening up opportunities in the fields of clinical diagnostic testing, human stem cell phenotyping, and in individual cell studies in biophysics [147]. Finally, microfluidic approaches seem to be the future for the mechanical analysis of cells.

### 3.5. Optical Magnetic Twisting Cytometry (OMTC)

The biophysical technique OMTC is based on magnetic fields to move microspheres (Figure 3). An external magnetic field is imposed to stress ligand-coated magnetic microspheres attached to specific membrane receptors. The mechanical characteristics of the cell can be inferred from the data obtained by measuring the resulting microsphere rotation [150]. Ferrimagnetic microspheres with a diameter of 4.5 µm, which are functionalized with the RGD peptide (the three amino acids arginine (R), glycine (G), and aspartic acid (D) are linearly arranged) or fibronectin, are added onto the cells and left for 15–20 min. To remove unattached magnetic microspheres, the cells are rinsed twice with PBS. In each experiment, the magnetic microspheres are magnetized horizontally and exposed to 60 cycles or more of rotation in a magnetic field oscillating at a constant frequency of 0.3 Hz. The system enables accurate synchronization of the magnetic field stimulus, which is applied to exert a controlled stress on the microspheres attached to the cell surface, and the acquisition of images to trace the displacement of the microspheres (for details see [151]). F-actin stiffness (G’) is calculated by division of the applied magnetic torque by the detected lateral microsphere displacement. Quantifying the displacements of the magnetic microspheres enables the calculation of the cell stiffness in units of pascals per nanometer, the area of the microspheres, and the cell complex modulus [131,152]. To guarantee precision, the determination of G’ is carried out six to twelve times at each experimental setting. The baseline cellular stiffness value is referred to as G’_0_. To ease the direct comparison of stiffness data between the various experimental runs and different groups, G’ values in every experiment are standardized to G’_0_. Thereby, different measurements form various analyses can be compared. A conventional way to distinguish the elastic response from the dissipative response of any kind of material is to determine the response to oscillatory loading and quantify the mechanical behavior by the complex elastic modulus. In particular, the technique permits the analysis of cell characteristics on micrometer–nanometer length scales (Table 3). In addition, it facilitates the simultaneous examination of a large quantity of cells. The beads themselves can be functionalized, whereby specific receptors and cellular structures can be characterized. Moreover, the size of the magnetic element can simply be upscaled. Overall, the technique is relatively easy to implement in the laboratory and is very cost-effective.

### 3.6. Discussion of the Culture and Measurement Conditions as Well as Selection of Cancer Cells

When using AFM measurements to determine the mechanical properties of individual cancer cells, the type of cantilever employed for the measurement and whether a sphere is glued to the cantilever play a role. In addition, the cantilever can be coated with proteins, peptides, or antibodies to probe specific receptors on the cell surface. In AFM measurements, the impact of cell adhesion can be addressed, whereas it cannot be explored using the optical cell stretching technique (Table 3). In optical cell stretching, the mechanical signals of the TME, such as ECM stiffness, are decoupled from the cellular mechanical properties, which can lead to changes in the cellular mechanical properties because bi-directional crosstalk is not possible [50,57,153,154]. As the malignant transformation of cancer has been attributed to alterations in both extracellular and intracellular mechanical characteristics, several studies have implied that these modified physical characteristics are instrumental in fostering cancer advancement [50,155,156,157]. Consequently, the stiffness of the cancer cells and the stiffness of the TME cannot be disentangled, as is the case for optical cell stretching measurements. This raises the question of whether these mechanical measurements are still cutting edge. In contrast to measurements with optical cell stretchers, AFM, MPA, and OMTC measurements can be performed by integrating the effect of the mechanical cues of the microenvironment.

In a rare investigation of the effects of culture conditions on mechanical stiffness measurements, it has been demonstrated that the composition of the growth medium influences the elasticity of MCF10A (normal controls) and MDA-MB-231 human breast cancer cells [158]. The average Young’s modulus of both cell lines dropped by 10–18% upon reducing the serum from 10% to 5% and supplementing the medium with 20 ng/mL epidermal growth factor. Although these modifications in elasticity could potentially have several biological implications, none of them were statistically reliable. When exposed to a medium supplemented with epidermal growth factor (EGF), cholera toxin, insulin, and hydrocortisone, the MCF10A cells exhibited a significantly greater change in elasticity compared with the MDA-MB-231 cells. All of these components are part of the recommended routine cultivation medium for MCF10A cells (M5) [158]. MCF10A cells cultured in M5 medium became significantly softer in comparison to a standard MDA-MB-231 medium (M1). Consequently, the cultivation conditions prior to and concurrent with the mechanical characterization of cancer cells may be critical for analyzing and comparing the mechanical properties of different types of cells. Thus, these conditions should be kept the same. Otherwise, differences in the softness/rigidity values may be obscured by the different cell reactions to culture media and their respective components. In line with these results, extremely important is the enzymatic detachment, usually referred to as trysinization, and resuspension of adherent cells in a suitable buffered medium or buffers. Instead of trypsin/EDTA, an Accutase solution is frequently used [159,160,161]. However, what sense is there in making cell stiffness measurements in buffers or of adherent cells in a non-adherent state? In addition, it needs to be considered that the differentiation of cells, such as cancer stem cells, can be altered by removing water from cells by increasing the osmotic pressure on them.

Another issue is the heating of the cell during the measurement procedure at high laser powers, since the mechanical properties are temperature dependent (Table 3) [162,163]. However, the laser powers are adapted to the measured cell types. For the capability to assess whole-cells, like in optical cell stretching, spheroidal cell deformation in a suspension minimizes variability resulting from the point of contact of AFM tips or micropipettes on mechanically heterogeneous attached cells such as significant stiffness variations in proximity to or distance from actin stress fibers [164]. For cases in which this heterogeneity is of relevance during adhesion [165], it is not possible to directly study the cells using optical cell stretching. Instead, when heterogeneity is required, it can be achieved using AFM and micropipettes (MPA). When high throughput analysis is required, microfluidic systems are suitable for detecting changes in subpopulations in various cancer cells due to their high throughput of cells to be analyzed. These systems can also be combined with other cell biological techniques, which only further emphasizes their importance in the field of mechanobiology in cancer research.

When determining the Young’s modulus, it needs to be considered that it is subject to change. Research revealed that the Young’s modulus of cells cultivated on soft substrates decreased when compared with cells cultivated on stiff materials [166]. Earlier investigations have demonstrated that the stiffness of cells rises with a growing stiffness of the substrate [167]. Moreover, the cellular Young’s modulus was found to initially grow and subsequently drop with rising stiffness of the substrate [168]. This finding is supported by the results of other in vitro investigations indicating that cell stiffness varies according to the substrate [169,170]. In addition, stiffening of the ECM caused by cancer advancement resulted in augmented motility of cancer cells [171]. These results suggest a tight connection between cell stiffness and cancer cell invasiveness, with the Young’s modulus potentially representing a suitable target for the impairment of the invasiveness of cancer cells.

Apart from the effects of the culture and measurement conditions, previous efforts to clarify the question of whether soft cancer are more malignant have concentrated on mechanical comparisons of cancer cells and normal cells [4,17,27,28,42]. These investigations have provided enormous insights into the part played by cellular mechanics in invasion and metastasis, and have given rise to the idea that cancer cells must be extremely deformable to pass through narrow tissue spaces [3,24]. While there are in fact multiple studies demonstrating that cancer cells are softer in comparison to normal cells [4,27,28,42], this result is not universally valid by far, as some reports have indicated that cancer cells are stiffer in comparison to normal cells [41,95], notably when they come into direct interaction with the ECM [46,172,173,174]. In addition, the differences in the genetic background of normal and tumorigenic culture models complicate the interpretation of many of these studies, so it is challenging to determine whether measured variations in cell stiffness are actually related to variations in tumorigenic potential. In addition, when mechanical comparisons have been carried out within isogenic tumor advancement, it was found that the cell stiffness rises with tumorigenic capacity [96].

## 4. Alternative Mechanical Analysis Approaches in Cancer

Since the methods and results presented thus far aimed to mechanically characterize the cells as a uniform entity, alternative approaches to the classical mechanical characterization of cells are now presented. These approaches also take into account the possibility that a single mechanical measurement of the cells at a specific point in time is not sufficient to characterize them mechanically. In addition to determining the mechanical properties of the entire cell, certain cell components, such as the cell membrane or the cell nucleus, are now also being mechanically characterized. There is also an increasing emphasis on a more physiological environment for the cells. These new directions in the mechanical characterization of cells are in line with the fact that cancer is linked to tissue stiffening evoked by the increased production of ECM and increased growth of the tumor. Therefore, it seems likely that cancer cells may not display the same softness/stiffness throughout the entire cancer evolution, instead, they may adapt to the specific mechanical requirements for each step of the metastatic cascade during the malignant progression of the tumor. For example, when cells are exposed to stress from simple flow, filamin accumulates throughout the cell and increases the overall mechanical stability of the cytoskeleton [175].

In the following, special emphasis is placed on the identification of other mechanical or morphological biomarkers or on the functional probing of the stiffness/softness of cancer cells such as cancer cell volume alterations in the volumetric compression of cancer cells, mechanical adaption, and patchiness.

### 4.1. Membrane Tension of Cancer Cells

The membrane tension analysis of cells can be used, as cells in tissues are exposed to a variety of forces that govern their fate and regulate development and homeostasis like tension homeostasis [176]. Cells mechanically perceive their surroundings via localized mechanoreceptors, which is referred to as mechanosensing [57], thereby enabling them to react to mechanical stimuli by adjusting their cytoskeletal framework or the structural organization of the plasma membrane [57], all of which seem to involve membrane tension alterations. For instance, cells mechanotranslate force and modify their phenotype by a process referred to as mechanotransduction. Cells adjust their tension when subjected to long-term force by activating cellular mechanisms that regulate actin tension, thereby promoting matrix reorganization and stiffening, and modifying cell–cell adhesions so that cells attain a condition of tension homeostasis [177]. The tension of the cell plasma membrane can be assessed by monitoring the relative depth of indentation at constant force when an AFM cantilever probe tip is pressed onto the cell surface [178,179].

The activity of oncogenes can lead to a deterioration in tensional homeostasis. In addition, the impairment of tensional homeostasis is also promoted during the advancement of cancers and the appearance of tissue fibrosis. Consequently, the disruption of tensional homeostasis is associated with heightened carcinogenic susceptibility. The mechanical stresses that arise in cancers can also facilitate mesenchymal transdifferentiation of cells to acquire a stem-cell-like trait that promotes their aggressiveness, metastatic distribution, and chemotherapeutic resistance. Therefore, developing strategies that enhance cancer mechanics could be an attractive approach to counteract the aggressive malignant behavior of the cancer. Impairment of tensional homeostasis can lead to an adhesion mode switching in breast cancer cells [176,180]. These results suggest that elevated membrane tension enhances the formation of clathrin-containing adhesion complexes (CCACs) at integrin αVβ5 adhesion sites, which results in the reduced proliferation, spreading, and migratory capacity of cancer cells. In contrast, a lower membrane tension encourages the development of focal adhesions, which coincides with the softer membranes found in cancer cells and therefore possibly fosters the advancement of cancer. Conversely to the mechanoregulatory influence of membrane tension on CCACs, focal adhesions are likewise controlled by fluctuations in actomyosin contractility across the cell cycle. Consistent with a morphogen-like mechanism, the cell cycle is interrupted through a periodic switching between contractile modes, fueled by the intracellular rearrangement of the actin cytoskeleton from cortical to stress fibers within N2A and CHO cells [181]. At the switch from the G1 to S phase, actin reorients from a mainly cortical organization into stress fibers and resumes a cortical organization near the end of the G2 phase in SUM159 breast cancer cells (Figure 4). This finding explains the increased membrane tension seen throughout early G1 and late G2 because cortical actin organization is understood to be a source of membrane tension [182,183]. Conversely, low membrane tension in the late G1 and S phase is linked to the emergence and stabilization of stress fibers, which fosters the generation of focal adhesions. Moreover, the disruption of cortical actin organization in G1 leaves actins more exposed, potentially promoting their engagement during clathrin-coated structure invagination [184], thus facilitating CCAC uncoupling.

To substantiate the hypothesis and identify a connection between membrane tension and the change in adhesion mode, chemical and mechanical strategies were utilized to ectopically alter the membrane tension. For example, fasudil, which is a Rho kinase inhibitor, has been found to elevate membrane tension and increase cortical actins. In turn, leptin, which enhances stress-fiber-driven actomyosin contractility, reduces the membrane tension [185]. These findings were confirmed by a comparative analysis of the phosphorylated ezrin, radixin, and moesin (ERM) proteins within these cells. pERM conveys the membrane–cortex linkage and is thus coupled to elevated membrane tension [186,187,188,189]. Concurrent with earlier results, treatment with fasudil caused a substantial rise in pERM translocation toward the plasma membrane. Inversely, leptin-treated cells displayed mainly cytoplasmic pERM without evidence of pERM aggregation at the plasma membrane. With the increased membrane tension caused by fasudil, a prevailing association of CCAC with integrin αVβ5 was evidenced. In contrast, leptin administration enhanced the engagement of αVβ5 sites specifically by focal adhesion complexes [176]. In summary, measurement of the tensional homeostasis seems to be suitable for identifying mechanically altered cancer cells during the malignant progression of cancer.

### 4.2. Cell Volume Alterations Due to Compression Can Serve as an Indicator for Malignancy

Cell stiffness may be altered easily by cell volume changes. Instead of cellular mechanical properties, the cell volume is proposed to be a well-regulated cellular characteristic and increases when the cell grows, whereby the cell maintains its macromolecular density. In reaction to extrinsic physical stimuli, the cell volume has been found to undergo alterations, causing water influx/outflow and substantial alterations in its subcellular macromolecular content density. For instance, when cells expand on a substrate, they decrease their volume and intensify their molecular compaction, which results in the drainage of water. Upon closer examination of this effect, it was found that water is drawn out of mesenchymal stem cells due to osmotic pressure, and this was enough to change their pathway of differentiation [190]. On the basis of these findings, it can be assumed that cells take different differentiation and functional routes by sensing/modifying their cytoplasmic volume and density by altering the water inflow/outflow [190]. In addition, the cell volume can change when cells pass through narrow constructions in a fast manner. In this context, Cl^−^ acts as an integral component of the osmotic regulator of cell volume and serves as the principal driving force for the volumetric alterations necessary for immature cells to proliferate and migrate [1]. However, the cell volume can also change on a much more rapid time scale, for example, on cell migration through confined spaces [191,192]. Confinement can cause the compression of cancer cells. Moreover, cellular compression promotes various transcriptomic and phenotypic adaptations in melanoma [193]. These two characteristics become apparent in a cancer setting as heterogeneity [157,165,194,195] along with both genetic and phenotypic plasticity [165,195,196]. During cancer development, melanoma cells can face a range of environmental hurdles that cause them to adapt. For instance, melanoma cells can take on a stem-cell-like characteristic and increase tumorigenicity at the edge of tumor tissues [197].

An important biophysical indication of cancer is volumetric compression [198]. The proliferation of cancer cells in confined tissue surroundings results in the compression of neighboring cells [199,200,201,202]. In addition, cancer cells can travel through dense ECMs, transmigrate through endothelial vascular linings and basement membranes into the vascular system, and extravasate into the interstitial tissue to form secondary tumor sites in the course of tumor dissemination and metastasis [203]. These incidents take place in restricted microenvironments with cross-sectional areas varying from 10 to >300 µm^2^ [204]. To enable metastatic spread, cancer cells must change their shape to pass through subnucleus-sized openings in the ECM [204]. The deformation of cancer cells [205] can result in a reduction in the cell volume [190,192]. On top of the solid stresses, the TME also experiences increased fluid pressure, which is capable of compressing cells [206,207]. In numerous investigations, hyperosmolarity has been examined as a biocompatible trigger for causing a controllable volumetric compression within diverse cell types, among them mesenchymal stem cells [190,208], adipocytes [209], breast cancer cells [198], lung cancer cells [210], cervical adenocarcinoma cells (HeLa Kyoto), and myeloid leukemia cells (HL-60/S4) [211]. In addition, volumetric compression, which can be triggered through both osmotic and mechanical compression, can increase intestinal stem cell self-renewal and enhance intestinal organoid generation via Wnt/β-catenin signaling activation, suggesting that both compression types trigger equivalent mechanotransduction reactions within cells [212]. These results once again confirm that cells are able to demonstrate phenotypic and genetic plasticity in the face of volumetric compression, which can be caused by various stimuli like osmotic pressure and mechanical compression.

There are two faces to this coin: it has been shown that compressed cells block the cell cycle, decrease the migratory phenotype, raise organelle pressure, and increase oxidative stress. In addition, compressed melanoma cells are able to withstand subsequent exposure to cisplatin better than non-compressed cells, resulting in improved survival of the cancer cells. Collectively, compression has been determined to have both tumor-promoting and tumor-suppressing actions on melanoma advancement [193]. Moreover, the nuclear translocation of the transcription factors Wnt/β-catenin and YAP/TAZ are key drivers of tumor motility and proliferation [213,214,215]. In contrast, a diminished nuclear translocation and cytoplasmic macromolecular crowding of both transcription factors has been observed, which could likewise be connected to the suppression of tumor advancement. Compression has emerged as a crucial physical factor that increases the resistance of cancer cells toward chemotherapy [193]. In summary, it can be assumed that volumetric compression exerts a dual function in melanoma advancement. On one side of the coin, compression impedes the proliferation and migratory behavior of melanoma cells, hence suppressing tumor advancement. Compression can also cause subcellular activities like endoplasmic reticulum (ER) stress, impaired mitochondrial functionality, and disruption of the nuclear envelope, which can lead to enhanced intracellular organelle stress and oxidative stress. The other side of the coin is that compression triggers transcriptomic activities like adjustments to membrane transporters, antioxidant reactions, and the cytoplasmic removal of toxins, which increases the survival rate of cancer cells after chemotherapy. Overall, the work revealed how melanoma cells adapt in response to physical stress, which could act as a natural selection mechanism governing the evolution of cancer cell subsets. The presented study highlights the physical stimulus of volumetric compression, particularly triggered by hyperosmotic stress, as one possibility for a mechanically-induced adjustment of the cell state. This leads to the question of whether the mechanical adaptations themselves are crucial for carcinogenesis and malignant progression, which is discussed in the following subsection.

### 4.3. Mechanical Adaption of Cancer Cells Is Required to Follow Their Metastatic Path

Cancer cells are often significantly softer in comparison to normal counterparts (see Table 1). This finding has led to the intriguing hypothesis that cancer cells adaptively soften in the course of invasion and metastasis [3,24]. Since metastatic cells are generally softer than their non-malignant equivalents, it is assumed that high deformability in both the overall cell and its nucleus provides a considerable benefit for metastatic capacity. Whether there is a delicately balanced, fixed mechanical state that encompasses all the mechanical attributes needed to survive across the entire cascade, or whether cancer cells must dynamically optimize their characteristics and intracellular components at every new phase, has been less explored. In this subsection, the various mechanical challenges that cancer cells have to successfully overcome along their metastatic cascade are examined, and the possible adaptation of the properties of cancer cells in response to these challenges is discussed [216]. In this process, the mechanical signature of a successful cancer cell could be its capacity to adapt to the consecutive restrictions of the microenvironment. In this context, there is compelling experimental evidence that cancer cells become softer when they move through confined spaces. It has long been suspected that targeting the softness of cancer cells to prevent their invasion and metastasis could aid in the overall treatment of cancer. However, it has yet to be directly proven that cancer cells become soft as they travel across narrow spaces. The unclear picture in the progression of cell stiffness throughout invasion is largely due to the absence of direct measurements of cell mechanics in the invasive process. This knowledge gap is certainly due partly to the technical difficulty of measuring the mechanical characteristics of a cell in the restricted migration. To close this gap, Rianna, Radmacher, and Kumar developed a microculture system that allowed them to study how the mechanical characteristics of cells evolve as they migrate in a confined space [34]. Their biophysical device consists of an open microfluidic system made of polydimethylsiloxane (PDMS) with topographically structured channels that are coated with fibronectin. They demonstrated that human U2OS osteosarcoma cells became significantly softer when they passed through the channel, whereby the Young’s modulus declined from 5.6 to 2.1 kPa. This drop in stiffness was coupled with an elevated actomyosin concentration at the side walls and the nuclear exclusion of Yes-associated protein (YAP) [34]. In the next step, the cytoarchitecture of the cells in the course of the transition from free to confined migration inside the channels was elucidated using integrated confocal microscopy of the actin cytoskeleton. In contrast, U2OS cells in non-confined geometric arrangements mainly formed stress fibers at their basal surface, exhibiting the anticipated pattern of fibers with dorsal and ventral fibers together with transverse arcs [217]. In contrast, cells within confined geometries positioned the stress fibers predominantly at the channel walls, giving rise to two parallel rows of actin bundles and few stress fibers juxtaposed to the bottom of the device. Whether these alterations in mechanics and cytoarchitecture are followed by functionally important mechanotransductive signaling processes has been addressed by assessing the localization of YAP, which has been reported to translocate to the nucleus upon exposure to increased cellular contractility [218,219]. As suggested, confocal imaging revealed the strong nuclear translocation of YAP for cells in non-confined topographies and the strong cytoplasmic accumulation of YAP for cells confined in channels [34]. Nevertheless, when cells made contact with one channel wall or when the cells were not fully enclosed [34], YAP was primarily found in the cell nuclei. This result is consistent with AFM measurements demonstrating that cells undergo softening when they pass through confined channels. In summary, cells in non-confined areas of the substrate are stiff, have well-developed basal stress fiber scaffolds, and exhibit strong YAP localization to the nucleus. When cells are subjected to growing constraints, they soften, enhancing the actin density along the channel walls and displacing YAP out of the nucleus into the cytoplasm. These outcomes are congruent with earlier findings that YAP’s nuclear translocation is tightly linked to actomyosin assembly and contractility [219], supporting YAP’s involvement in mechanotransduction [218,220,221]. In previous works, cell stretching uncoupled from topographical constraints was identified as a second mechanism contributing to altered cytoarchitecture and YAP localization [222,223]. When cells were placed on 1D microlines, YAP persisted in the nucleus, suggesting that cell softening and YAP function per se rely on topographical confinement and are not merely controlled by alterations in cellular extension. In future efforts, it would be interesting to determine the tensile forces at the channel walls and explore how the distribution of the stress fibers relates to where YAP localizes. Moreover, it would be interesting to analyze which YAP-dependent genes are most severely impacted by the geometry of the confinement and to which degree the subsequent gene products are involved in constricted motility.

Importantly, when confined by culture on narrow, 1D ECM microlines, the cells elongated without softening, suggesting that cell softening occurs specifically in 3D geometries. In addition, AFM indentation can be used to track alterations in cell mechanics prior to and subsequent to constriction. To ease interpretation, each cell can be categorized into one of four groups: (1) not confined; (2) connected to one wall; (3) connected to two walls but not completely confined; and (4) completely confined. A comparative analysis of the cells in every group revealed that U2OS cells became increasingly softer as they transitioned from unrestricted migration to semi-restricted migration to fully restricted migration [34]. These data provide the clearest evidence thus far that cancer cells soften during restricted migration and lend weight to the notion that the softening of cells during invasion is a mechanoadaptive mechanism. The weakness of the study, nevertheless, is that only one type of cancer cell, the human U2OS osteosarcoma cells, was examined and that the ECM (e.g., fibronectin) was not heterogeneous. Nevertheless, these findings confirm the hypothesis that cancer cells undergo a transition from stiff to soft cells. This mechanical transition seems to be related to the classical EMT transition of cancer cells during the malignant progression of cancers. This raises the important question of whether the transition is permanent or temporary. In particular, it must first be examined whether stiff cells also remain stiff over time, and second, whether this cell stiffening is independent of environmental influences. The last part of the question can be answered with “no”, since cellular mechanics like stiffness depend on environmental influences.

The new biophysical technique will provide additional knowledge on the question of whether softer cells are more mobile and aggressive by mechanically characterizing cancer cells at different levels of confinement and invasion. It has been revealed that cancer cells that are exposed to increasing spatial constraints become significantly softer, which is congruent with the notion that deformability aids invasion and metastasis. These findings are in agreement with two recent investigations. The first examined the traction forces of cells within confined microenvironments and revealed that 3D restriction decreased the traction forces of cells [224]. The second investigated the migration of breast cancer cells through collagen so-called microtracks and revealed that this restricted geometry decreased both the basal focal adhesions and stress fibers. Specifically, the latter work revealed that confinement also enhanced matrix strain, which both adds to and contrasts with the AFM stiffness measurements discussed above, and suggests that stiffness and contractility are not fully connected [225].

In summary, this biophysical technique allows for the investigation of cell motility from the non-confined to the confined environment, along with the assessment of the mechanical characteristics of the cell at the supernuclear area [34]. The system configuration also turned out to work very well for high-resolution optical microscopy, which also comprises super-resolution imaging. These studies found that cells in unconstrained geometries took on a highly extended phenotype with typical stress fiber arrangements at the cell base and spanning over the nucleus. When cells become more and more constricted, they are more stretched out and display elevated actin concentrations at the lateral walls of the device, which has also been seen in microchannel analyses [226], but could at that time not be attributed to a specific function. This increased lateral actin organization under confinement may be to encourage restricted migration by using the walls as “handles” to generate traction and migrate while also lowering their cortical stiffness. Importantly, cortical stiffness measurements, as obtained by AFM, do not equate to measurements of traction force or contractility, although the correlation between the two is usually considerable [227,228]. It seems probable that the contradictory findings regarding how stiffness alters during carcinogenesis, in that cells appear to become more compliant and produce larger contractile forces when constrained, thereby exhibiting both increased and decreased stiffness, can be attributed to the measurement approach.

Despite the promising results of the new biophysical technique, there are still some frontiers that need to be overcome. For instance, it could be useful to quantify the variations of the viscous (loss) cell modulus due to confinement as a supplement to the elastic properties that are frequently determined. Moreover, it would be instructive to see whether there is any interesting correlation between the softening caused by the constraint and malignant behavior, for instance, in a series of progressions with carefully matched genetic backgrounds. Ultimately, it is essential to verify whether the findings also apply to models that more closely resemble actual tissue such as organoids and animal models. Moreover, immune cells are extremely susceptible to the mechanical characteristics of cancer cells. Apart from the intrinsic advantages of softening cancer cells throughout the malignant progression of cancers, soft cancer cells can escape immune-mediated elimination [60]. Consequently, targeting this mechanical immune checkpoint could improve the potency of cancer immunotherapy.

### 4.4. Patchiness (Local Heterogeneity) as a Marker of Malignent Cancer Cells Versus Stiffness

While the mechanical phenotyping of various cancer cells at the single-cell or population scale represents traction and softness/stiffness as emerging diagnostic tools, it is still uncertain whether these measurements can provide diagnostic and prognostic insights. Several lines of evidence have pointed out that cell stiffness can influence their metastatic capacity, whereby highly invasive malignant cells tend to be softer on average (at the population level) compared with less invasive cells [24,44] (see Table 1). As no two individuals are exactly alike, measurements of cell stiffness yield a cell stiffness distribution for each cell population, regardless of the method of measurement. Such a distribution results from phenotypic heterogeneity at the cellular level and can endure for several cell divisions [12,22,158]. Whereas the heterogeneity of the mechanical characteristics of cells (like cell stiffness or adhesion) has been identified and reported earlier, the mean values of these distributions are employed to discriminate the two cell types and possibly to evaluate the malignancy and metastatic capacity of a certain cell population [4,22,37,158]. The distribution of the mechanical characteristics of cells within a cell population, whether it is a tumor or a normal tissue, can yield valuable information about how the particular population will progress with time and possibly evolve into a malignant state and form metastases. The rationale for this approach is that even healthy cell populations exhibit a wide distribution of cellular mechanical characteristics, pointing to the existence of at minimum a limited number of cells with cancer-like phenotypes at the ends of these distributional ranges [12,15,22,24,158,229]. In addition, the in situ mechanical characterization of solid tumors at the micrometer scale demonstrated that tumors consist of areas of soft and stiff cells [23], and that the definition and arrangement of these areas reflect tumor malignancy [230]. Consequently, computer simulations of cell–cell interactions in a 3D tissue environment have been performed, and the development of the cell population exhibiting heterogeneous mechanical characteristics is followed over time. Focusing on cell stiffness as an important mechanical characteristic that is heterogeneous between individual cells, a targeted search has been conducted for the growth of cells with lower stiffness (softer cells) because this is a cell trait that is closely linked to cancer [12,22,231].

In subsections, the heterogeneity of the mechanical characteristics of cells originating from a single biological source, like a tissue or a solid tumor, is considered as a candidate for a possible new biomarker. In the past, the heterogeneity of cell mechanics has generally been underestimated in mechanobiology. Mechanics-based in silico models of cell–cell interactions and cell population dynamics in 3D environments have been employed to investigate how the heterogeneity in cell mechanics affects the dynamics of tissues and tumors. These simulations reveal that the initial heterogeneity in the mechanical characteristics of individual cells and the organization of these heterogeneous subgroups within the microenvironment can determine the dynamics of the entire cell pool and drive a transition that favors the proliferation of malignant cell phenotypes in healthy tissue contexts. The global heterogeneity in the cellular mechanophenotype and its distribution in space is described by a “patchiness” index, which reflects the ratio of global to local heterogeneity within cell pools. There is a threshold above which a cell population that is healthy overall begins to exhibit a continuous transition into a more malignant phenotype. These results have led to the hypothesis that the degree of “patchiness” of a tumor or tissue specimen can be an important early predictor of malignant transformation and cancer onset in benign tumors or normal tissues. Moreover, it has been speculated that the irregularity of tissues, as quantified by either biochemical or biophysical signatures, could become as important a measure for predicting tissue health and disease proneness as the irregularity of topography is in ecological systems [232]. Apart from patchiness, there is another phenomenon that has not been addressed widely, which is cancer cell clustering. It has not yet been addressed when the stiffness is determined at the individual cell level. However, bulk measurements of the entire tumor are not helpful, since then the heterogeneity of the cells is not kept within the cluster as the cells in a cluster are not synchronized in the cell cycle, and different cell cluster may exhibit unequal proliferation rates. Consequently, the mechanical properties of individual cells and their location in the tissue are critical to analyze. Although rare, understanding the interplay between mechanical fluctuation and spatial clustering, and how it can promote the growth of stiffer cells, can be beneficial. It may help to understand why there is an outgrowth of stiffer cancer cells when the tumor progresses. For example, the probability of a tumor arising increases when both the mechanical variation and the accumulation of cells with similar mechanical stiffness in a cell cluster in an originally healthy tissue grows [232].

How can cells survive when they need to resist forces? The necessity for the spatial clustering of cells with analogous mechanical characteristics for tumorigenesis highlights the interdependency of the interactions between tumor-like cells required to foster their enhanced proliferation. Perhaps this can be attributed to the increased surface area of tumor-like cell clusters in the tissue, thereby conferring on them the capacity to withstand the force exerted by the encircling stiff cells [233]. This can be linked to experimental observations that cancer cells are capable of persisting in their stiff surroundings as they multiply and expand, propelled by elevated homeostatic pressures for these cells [234,235]. Using the patchiness index for the tissue, the impact of the total variance (heterogeneity) in the cell mechanophenotype and the local clustering of mechanically analogous cells has been evaluated. Regardless of the cubic window size employed to calculate the patchiness index (3 cell lengths, 4 cell lengths, or 5 cell lengths along each dimension), tissues with a patchiness index exceeding 0.85 have been found to have a significant probability of exhibiting a reduction in mean cell stiffness and a concomitant malignant mechanophenotype shift [232].

What are the weaknesses of the patchiness model? An entirely theoretical approach was employed to investigate the development of a mechanically diversified population of cancer cells in a tightly compacted 3D tissue setting. Excluding the occurrence of nutrient gradients, ECM, or considerably distinct cell types, the focus has solely been on epithelial or near-epithelial cell types. It is presupposed that cell–cell crosstalk is strictly mechanical and that cell behavior decisions are also mainly driven by mechanosensitive back-coupling based on observational experimental evidence. Although all of this seems to oversimplify a complicated tissue system, these types of simplified models have been utilized to comprehend and characterize processes associated with tissue [236,237], tissue growth [238], cancer incidence [233], cancer growth, and metastatic dissemination [239,240,241]. Moreover, the limited length scale of the tissue model simulated in this instance permits the hypothesis of a dense aggregation of tightly arranged cells excluding the presence of an ECM, gradients of nutrients, or vessels. In the future, it is possible to expand the model and include many more details.

What about the macroscale heterogeneities? Notably, as mentioned earlier, this simulation study concentrated on individual-cell-scale mechanical heterogeneity and patchiness on a few-cell of length scales such as about 40 μm. The length scale is pertinent to recent work characterizing tissue mechanophenotypes [23,97] and in following the effects of stiffness alterations on single-cell fates in tissues and cancers. The emergence of mechanical heterogeneity within cell populations at this length scale can be explained by the random genetic mutations, gene methylation patterns, or phenotypic alterations resulting from local environmental features such as ECM characteristics and nutrient supply. Nevertheless, this model fails to address questions concerning macroscale heterogeneities within tissues and contributing factors to whole-tissue stiffness readouts, like the attraction of several other cell types, including fibroblasts and endothelial cells, and of cellular structures such as vessels as well as immune cells including neutrophils, T lymphocytes, macrophages, and NK cells into the TME, and the resulting changes in the stromal tissue enveloping solid cancers.

In the context of the limitations outlined above, the model can be utilized to assess how the heterogeneity of the cancer cell mechanical characteristics, particularly cell stiffness as monitored and quantitated using experimental devices, can be exploited to foresee the emergence of tumorigenic growth and malignant progression in a supposedly healthy population of cells. Heterogeneity at the cellular population scale but homogeneity at the local scale, as characterized by the inhomogeneity of cellular distribution throughout the tissue system, indicates an enhanced probability of cancer onset and an overarching malignant skewing of an originally normal tissue. Importantly, high heterogeneity is not deleterious for tissue destiny per se, but only when the population stays well-mixed. Nevertheless, the accumulation of similar cell populations at a local level, which may arise from an underlying heterogeneity in the extracellular environment, the supply of nutrients, random cell division occurrences, or migration-dependent differentiation of cells, promotes malignant transformation in highly heterogeneous tissue types [242,243,244]. These findings are indeed restricted to the tissue-level heterogeneity on short length scales of 10 to 100 µm and disregard the larger heterogeneity of the tumor surroundings, which is connected to the agglomeration of various cell types within the tumor stroma as well as the level of vascularization and perfusion of nutrients. As the primary emphasis of this debate is on the transformation from a normal or a premalignant condition to a malignant one, the macroscale heterogeneities that are often seen in the vicinity of cancers are not considered to be of primary importance. In contrast, heterogeneity at the scale of tissue cell populations, which are maintained in homeostasis with their surroundings, is of greater importance for malignant transformation. Moreover, the mixture of cancer cells with other normal or tumor-associated cell types keeps them in a less aggressive state.

From the perspective of cancer ecology, these findings add to several ideas put forward previously [245,246]. For instance, high diversity within the cell population, which is characterized by a high Evo index, or the presence of a proper environmental niche, which is characterized by a high Eco index, have been postulated as ecological parameters that can predict tumor advancement, the progression of disease, and response to treatment [247]. The model presented investigates the contribution of both the general heterogeneity of the endogenous tissue system and the generation of suitable niche environments. In this case, local niches are determined by cell–cell instead of cell–matrix or cell–environment crosstalk, which is in line with the control of cell destiny in localized ecological niches through cell–cell interactions [248]. Nevertheless, the integration of additional governance driven by extracellular signals such as gradients of nutrients and oxygen, the availability of cytotoxic and genotoxic compounds, and direct cell–ECM crosstalk with the current mechanosensitive cell–cell interaction into future modeling efforts is conceivable [249,250]. An additional relevant aspect of both ecology and cancers, that of homeostatic competition [251,252], is also accounted for in the presented model. Homeostatic competition means a constant population density at the global level, which is governed through competition for resources or cooperation among the various kinds of species inhabiting an ecological habitat. Homeostatic competition posits that extreme cancer cell types are held in restraint by their neighbors with more normal phenotypes. When they clump together, extreme cancerous cell types can suppress cell death, outcompete neighboring cells for available space, and become the dominant cell subtype (Figure 5). Is this phenomenon jamming? Maintaining homeostasis on the tissue scale represents a key requirement built into the presented model of healthy tissue. The low-diversity tissue preserves homeostasis, with competition among cell types for room to grow. In this model, homeostasis is sustained by a tissue with strong diversity and mixing, so that the extremely cancerous cell types are constrained within the context of their more normal neighbors. When enough cancerous cells accumulate, nevertheless, their interactions enhance proliferation and attenuate apoptosis, enabling them to trump their neighbors for spatial resources and start imposing their dominance on the tissue. There is also agreement with the similarities between dormancy, the avenue effect, and growth retardation in ecological and cancer colonies [253]. Is the onset of cancer associated with a transition from a homogeneous to a heterogeneous population? In this context, the question has arisen as to whether the transition from a mixed to segregated population with regions of low heterogeneity is linked to the malignant progression of cancer.

Analysis of the spatial heterogeneity of TME has revealed additional parallels between ecological habitats and cancers. TME features not only rich diversity in the pattern of its native tissue cells, but also diversity in the cells of the encompassing stroma, specific immune cells linked to the tumor, and the global arrangement and orchestration of these cell types within a fibrous ECM [254]. From these large-scale tumor surveillance studies, several measures of spatial heterogeneity have been introduced, both parametric and nonparametric, to gain diagnostic and prognostic knowledge about the tumor [255,256,257]. Measures of this kind have yielded good results, particularly for advanced cancers, although they are not as useful in assessing the risk of benign neoplasms becoming cancerous or low-grade tumors turning into high-grade tumors. An important problem in oncology is the issue of why and when benign neoplasms, which are much more frequent in all life systems, even in mammals, turn into aggressive cancers that tend to expand and disseminate [258]. The shift from a homogeneous to a heterogeneous population (that is unavoidable in the majority of biological models) and the shift from a mixed to a separated population with areas of low patchiness (that is most probably the rate-limiting step), could be crucial to address this problem. The patchiness index can also contribute to establishing more effective parallels between cancer and ecological systems with a high number of neoplasms but without catastrophic events that throw the ecosystem out of balance (Figure 5).

Why have the majority of studies focused solely on cell stiffness? Why have they not also considered the cell adhesion strength, cell contractility, and nuclear deformability? The study by Mirzakhel and colleagues was limited to the heterogeneity of the cell mechanotype, which is characterized through cell stiffness, as stiffness is among the key mechanical characteristics that are recognized to differentiate cancer cells from normal cell populations and can also contribute to the classification of cancer cell populations according to their level of aggressiveness [24,44,259,260]. Moreover, instruments are currently being engineered that will enable the mechanical stiffness of cells to be analyzed and imaged in situ within tissue specimens [261,262]. Nevertheless, other cellular mechanical characteristics, like cell adhesion strength, cell contractility, and nuclear deformability, also reveal comparable levels of heterogeneity within tissue and cancer cell populations [263,264,265]. Both cell adhesion strength and cell contractility impact cell organization, form, and size within vertex-based cell–cell engagement models [266] in similar ways to cell stiffness, and it is anticipated that they will yield similar results to those of Mirzakhel and colleagues [232]. At present, there are limited models that consider the stiffness of the nucleus when examining cell–cell crosstalk and tissue dynamics, and this is an area that requires further exploration in the near future. Finally, the mechanical variations between cells emerge from biochemical disparities, which are probably caused by variations in the gene expression patterns in a cell population. Therefore, the phenotypic variations between cells can be mapped directly onto the mechanotypic variations among cells, and tissue patchiness assessment can be expanded to any spatial characterization of cells inside a tissue context. In fact, with the development of precise single-cell gene expression profiling and cellular tracking approaches, along with broad-scale spatial phenotypic and the mechanical characterization of cells in situ [267,268], it seems conceivable that tissues could be classified experimentally according to their patchiness in both mechanical and biochemical traits. In summary, it can be said that tissue patchiness appears to deliver key diagnostic and prognostic knowledge for malignant growth in normal tissue and benign tumors.

What about the mechanical stiffness alterations over time and spatial mechanical stiffness analysis? Heterogeneity can be considered a standard in biology and cannot be eliminated. It holds for all orders of magnitude and all species. In contrast, restricting population segregation and local niche generation could be a possible mechanism to prevent an equilibrium shift toward subpopulation dominance. An uneven distribution can be avoided by limiting factors that favor the proliferation of specific subpopulations alone or limiting factors that restrict cell migration in specific tissue areas. Biological processes and biochemical or biomechanical determinants that possibly enhance or attenuate patchiness in normal tissues or cancers and their relationship to real tumor growth and aggressiveness are topics that require additional study and debate. Nevertheless, the concept of patchiness could be applied to cancer ecology in the same way as it is used in the field such as evolutionary ecology and dynamics in populations. In the future, a therapeutic approach to minimizing heterogeneity could be to limit segregation of the population and prevent accumulation in local niches. A therapeutic approach for reducing patchiness could be the restriction of factors that favor the proliferation of only certain subpopulations or factors that impair the migration and invasion of cells in certain areas of the tissue.

## 5. Discussion of Whether the Mechanics of Cancer Cells Are a Cause or an Effect of Cancer Development and What Role They Play in the Malignancy

As cancer cells can adapt the metastatic niche’s tissue characteristics, it seems plausible to suggest that the mechanical properties of cancer cells are a prerequisite for the malignant progression of cancers. For instance, cancer cells need to deform themselves to pass narrow constrictions, ECM scaffolds, or cell layers, such as endothelial cells, all of which require specific mechanical characteristics of malignant and invasive cancer cells. Thus, the mechanical properties of cancer cells seem to be rather a prerequisite of the malignant progression of cancer than a consequence of the malignancy of cancers. In addition, the mechanical phenotype of cancer cells may not be purely static and could be subject to mechanical adaptation. Moreover, there is not just one migration mode, which may also require different mechanical characteristics for different migration modes. For instance, tracks may allow stiffer cells to pass, adhesion strength may play a role, contractile forces are discussed controversially, and the degradation of ECM and its deposition may favor or hinder migration. The mechanical characteristics of a living cell is crucial in a multitude of cellular behaviors and performances [269,270,271]. Cells can react to an augmenting stiffness of the ECM with the exertion of actomyosin-based contractile forces [169,272]. An endogenous contraction of this kind itself can enhance the cell stiffness [273]. Thus, these findings support the idea of a mutual interaction between cancer cells and their microenvironment regarding the mechanical properties [50,165,274,275,276]. To correctly perceive and react to the mechanical stimuli of the surroundings, the stiffness of a cell should also equal that of the ECM [169,277,278], indicating that soft cells are likely to persist in a soft stroma, and stiff cells will perform best in a stiff niche. Supporting this, soft 3D fibrin matrices were found to increase H3K9 demethylation and Sox2 expression and the self-renewal of melanoma stem cells, while stiff matrices had adverse impacts [279]. In line with these findings, cells with differing levels of stiffness may exist side by side in identical tumor tissue because of the heterogeneity of the mechanical microenvironments of the primary tumor [229,280].

Even with this insight, it is still not possible to directly demonstrate that cellular softness acts as a fundamental characteristic for tumorigenic cells. The overall question is whether there exists a functional relationship between cancer cell softness/stiffness and cancer development and malignant progression. An approach to address this question has been the sorting of mechanically altered tumorigenic cells. Therefore, a technique has been generated to successfully sort and identify tumorigenic cells. The task of identifying and separating highly tumorigenic and metastatic cancer cells within a heterogeneous cellular population is a formidable task. More specifically, microfluidic devices have been employed to sort marker-based heterogeneous cancer stem cells (CSCs) in mechanically stiff and soft subsets. It was conclusively verified that these soft cells are extremely tumorigenic and are capable of metastasizing [63].

The isolated soft cancer cells (under 400 Pa) but not the stiff cancer cells (over 700 Pa) from CD133^−^, ALDH1^+^, or side population-enriched CSCs are capable of initiating tumors when engrafted at a low number of 100 cells into NOD-SCID (severe combined immunodeficient) or immunocompetent mice [63]. CSC markers like CD133 and ALDH1 alone cannot provide a reliable prognosis for patients, as there are contradictory findings. On the one hand, CSC markers are associated with a poorer prognosis and poorer general survival [281,282,283], while on the other hand, evidence from some studies indicates that CSC markers are not prognostically relevant [284,285,286,287]. These contradictory results could be due to the assumption that marker-based CSCs comprise both soft and stiff CSCs. Therefore, cell softness can potentially serve as a physical marker that can be measured to assess the prognosis of cancer patients. This hypothesis is supported by the finding that the Wnt signaling protein BCL9L is elevated in the cells of soft cancer cells and modulates their stemness and tumorigenic capacity. In clinical specimens, BCL9L expression is associated with poor patient prognosis.

It is therefore likely that cancer cells have no static mechanical characteristics such as softness/rigidity, but rather that they adapt dynamically to the conditions of their environment or that the mechanical characteristics arise from the interaction with their environment. A positive association between enhanced tissue stiffness and aggressive cancer performance led to the development of a new model of cancer progression that relies on the static or dynamic stiffening of tumor tissue [288,289,290]. As a consequence, these findings give rise to the hypothesis that the intrinsic softness of cancer cells might act as a hallmark for highly malignant and metastatic cancer cells. Consequently, these results support the hypothesis that malignant cancer cells must adapt their mechanical characteristics, such as softness, before they can migrate and form metastases. These results do not support the assumption that even stiff cancer cells, such as breast cancer cells or melanoma cells, migrate and form metastases, which seems to be in contrast to the previously mentioned findings in Table 2. Based on these functional assays, it can be said that the softness of cancer cells is a prerequisite for metastasis in human primary MP1 melanoma cells, mouse 4T1 breast cancer cells, human MCF-7 breast cancer cells, and mouse B16 melanoma cells. The universal applicability of the findings is still questionable, as only two different cancer cell types were analyzed. The study results showed a weakness, as the metastasis potential of soft tissue cancer cells was only analyzed in a mouse model.

There is, nevertheless, still no evidence that the cells emerging from the stiffened tumor stroma are cancerogenic and metastatic cancer cells. In addition, there is uncertainty regarding whether the tumorigenic cells are stiff or soft. Regardless of the global stiffness, the local microenvironments are very heterogeneous for tumor stiffness [229]. Enhanced tissue stiffness can be ascribed to more ECMs, probably constraining the circulation of blood vessels and resulting in cancer hypoxia, which is a frequent feature of TMEs. In primary tumors, hypoxia is linked to the increased dissemination of metastases and worse prognosis in cancer patients [291,292,293,294]. Hypoxia fosters the development of human breast tumor repopulating cells [295], whereby hypoxic sites can be extremely soft as a result of local tissue necrosis and the breakdown of the ECM scaffold. Therefore, elevated tissue stiffness can lead to an increased number of soft cancer cells accumulating at hypoxic sites, favoring aggressive cancer progression. Concordant with these findings, cancer cells with the highest migratory and invasive capacity are 5-fold less stiff compared with those with the lowest migratory and invasive capacity [44]. These data strongly imply that only naturally soft cancer cells can be highly tumorigenic and metastatic within animal models. These observations are in line with biophysical studies that showed that metastatic cancer cells are markedly softer compared with non-metastatic cancer cells [3,22,24,229]. The softening of cancer cells is also congruent with the finding that cancer cells with low adhesiveness are prone to migration and the formation of metastases [296]. It is possible that on top of hypoxia, the stiffening of tumor tissue drives EMT, and that the stiffening of cancer cells of a specific subpopulation promotes the differentiation and migration of soft undifferentiated cancer cells out of the tumor stroma [279].

In the heterogeneous architecture of cancer tissue, a hard-to-detect subpopulation of stem-cell-like cells has been found that is associated with both relapse and metastases [297,298]. Employing engineered ECMs, geometric features at the edge of cancerous tissue were found to shape a cell population with stem-cell-like characteristics. These cells exhibit CSC characteristics in vitro and enhanced cancerogenicity in mouse models of tumorigenesis and lung metastases. Apart from cancer cells and classical CSCs, these stem-like cells may play a role in the malignant progression of cancers. Thereby, the interfacial geometry can alter the shape and adhesion strength through integrin α5β1, MAPK, and STAT activity, and signal transduction pathways such as the induction of pluripotency signal transduction in stem-like cells. These findings for several human cancer cell lines imply that interfacial geometry orchestrates a general mechanism for governing the state of cancer cells [197]. In a manner analogous to how a growing cancer can take over normal soluble signaling routes [157], these results indicate how cancer can also leverage geometry to coordinate oncogenesis. The interfacial geometry requires additional future research to probe for CSCs or cancer cells of various cancer types and at distinct cancer stages.

There is also another approach to create various stiff prostate cancer cells. The aggressiveness of prostate cancer is linked to cell-mechanical alterations that promote extensive cell deformation, which is necessary for metastatic spread. Stiff and soft cancer subtypes of prostate cancer patients have been identified based on the membrane tension [299]. Subtypes of solid and soft cancer cells were delineated based on eight genes associated with membrane tension using lasso regression and non-negative matrix factorization analysis to biochemical relapse and compared with patients with the “soft” subtype, whereby the finding was externally confirmed in three additional cohorts [299]. The ten most frequent mutation genes between the “stiff” and “soft” subtypes were DNAH, NYNRIN, PTCHD4, WNK1, ARFGEF1, HRAS, ARHGEF2, MYOM1, ITGB6, and CPS1. There was a strong accumulation of E2F targets, base excision damage repair, and Notch signal transduction in the stiff subtype. The stiff subtype exhibited significantly increased tumor mutual burden and T cell follicle helper cell levels compared with the soft subtype, along with the expression of CTLA4, CD276, CD47, and TNFRSF25.

## 6. Conclusions and Future Directions

Apart from the simple analysis of the softness or stiffness of cancer cells, which may rather affect the aggressiveness of cancer cells and metastasis in a cancer type-specific manner than in a universal manner, other factors such as cell–cell interactions, cell–matrix interactions, interactions with other cell types such as immune cells, endothelial cells, and stromal cells, and complex networks of the ECM scaffold may also be responsible for the progression of cancers. Beyond that, the discussion will focus on whether it makes sense to measure the mechanics of cancer cells such as the softness/rigidity of individual cells or cells in a cell cluster. Nevertheless, it will be necessary in the future to thoroughly determine whether the soft or stiff cancer cells show a metastatic and tumorigenic capacity in human patients. Combining the mechanical properties of cancer cells with biological markers of cancer malignancy could lead to a breakthrough in prognosis and therapy, since the latter can be individually adapted to the patient according to the malignancy prognosis. This approach is supported by the important finding that BCL9L has been identified as a biological marker for soft cancer cells, enabling them to be distinguished from their stiff counterparts. The existence of unambiguous biological markers for soft cancer cells would render future mechanical analyses of cancer cells largely superfluous. Nevertheless, such analyses are useful for understanding the physical aspects of the tumor metastasis process. Besides the traditional single-cell analysis of tumors, the examination of tumor tissue samples is promising because it can determine the patchiness, such as the global and local heterogeneity of the tumor tissue. The heterogeneity of cancers can occur at various length scales, such as genetic heterogeneity, epigenetic heterogeneity, transcriptional heterogeneity, and tumor microenvironmental heterogeneity including immune and stromal cell as well as mechanical heterogeneity [165,300], which can still pose a challenge to cancer comprehension in future analysis. Tissue patchiness could serve as an excellent marker for determining the malignancy of tumors. In particular, it could be used to better understand the transition from healthy tissue to tumor tissue and from benign tumors to malignant ones.

## Figures and Tables

**Figure 1 cells-14-00584-f001:**
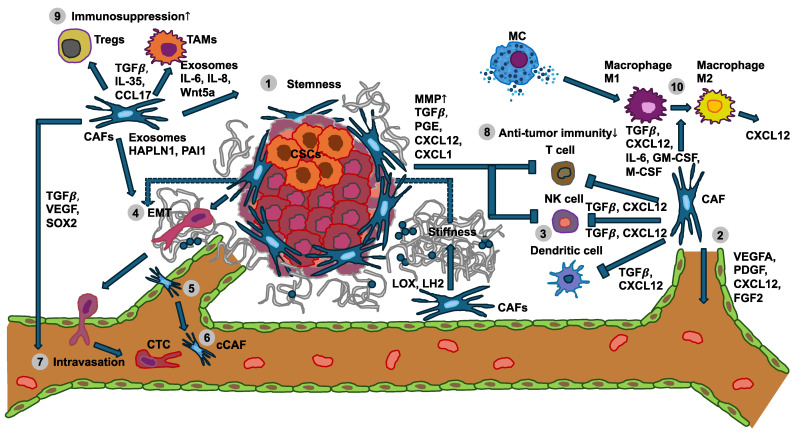
Interplay between the mechanical properties of cancer cells with immune cells and stromal cells like cancer-associated fibroblasts (CAFs) in cancer development and metastasis. (1) Various specific subtypes of CAFs release exosomes, interleukins (ILs), and Wnt5a to foster cancer stemness. (2) CAFs express and secrete soluble factors, among them VEGFA, CXCL12, FGF2, and PDGF, whose engagement with their receptors on endothelial cells leads to angiogenesis. Conversely, CAFs release LOX and LH2 to increase ECM stiffness, thereby facilitating VEGF/VEGFR crosstalk through integrins. (3) The tumor growth factor-β (TGF-β) secreted from CAFs can affect the differentiation and proliferation of T cells, attenuate MHC-facilitated immune identification, and decrease T cell penetration by enhancing matrix stiffness. In addition, TGF-β can also inhibit dendritic cells and the generation of cytolytic natural killer (NK) cells. The CXCL12 released by the CAFs can decrease the migration of T cells and suppress the proliferation of NK cells. (4) Several factors secreted by CAFs can increase the migration and invasion of cancer cells through the inducement of epithelial-to-mesenchymal transition (EMT). Moreover, CAF-enhanced matrix stiffness can also facilitate EMT. (5) CAFs regulate the intravasation of blood vessels through TGF-β, VEGF, and SOX2. (6) A cCAF subtype is implicated in blood vessel-associated metastasis and is characterized by CD44. (7) Metastasis-associated cytokines and exosomes released by CAFs in the primary tumor foster the generation of remote polymorphonuclear neutrophils (PMNs). (8) Anti-tumor immunity of NK cells, T cells, and dendritic cells is diminished by CAFs and the primary tumor. (9) Immunosuppression controlled by Tregs and TAMs is increased (indicated by the up arrow) through CAFs and the primary tumor. (10) CAFs and mast cells (MCs) can promote the transition of M1 macrophages to M2 macrophages. Cancer-associated fibroblasts (CAFs), cancer stem cells (CSCs), circulating tumor cells (CTCs), circulating CAFs (cCAFs), CXC-chemokine ligand 12 (CXCL12), epithelial-to-mesenchymal transition (EMT), fibroblast growth factor 2 (FGF2), hyaluronan and proteoglycan link protein 1 (HAPLN-1), interleukin 6 (IL-6); interleukin 8 (IL-8), lysyl hydroxylase 2 (LH2), lysyl oxidase (LOX), mast cells (MCs), natural killer cell (NK cell), platelet-derived growth factor (PDGF), plasminogen activator inhibitor-1 (PAI-1), polymorphonuclear neutrophils (PMNs), SRY-Box transcription factor 2 (SOX2), tumor associated macrophages (TAMs), tumor growth factor-β (TGF-β), vascular endothelial growth factor A (VEGFA), and Wnt family member 5a (Wnt5a).

**Figure 2 cells-14-00584-f002:**
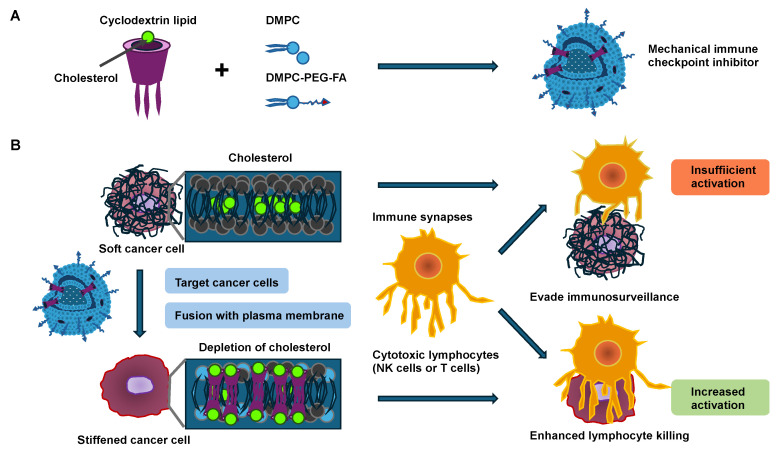
Mechanical immune checkpoint inhibitors (MICIs) cause the stiffening of cancer cells and increase NK- and T-cell-driven cytotoxicity. (**A**) After the synthesis, cyclodextrin lipid compromises three alkyl tails that are embedded as a nanostructure-element of MICIs. (**B**) Schematic representation of MICIs fusing with the cancer cell plasma membrane, thereby stiffening cancer cells by cholesterol depletion and exposing cancer cells to cytotoxic lymphocytes such as NK and T cells. Cytotoxic lymphocytes are more engaged with stiffer cancer cells, resulting in increased killing activity of the lymphocytes, whereas they are less engaged with softer cancer cells, resulting in immune surveillance evasion.

**Figure 3 cells-14-00584-f003:**
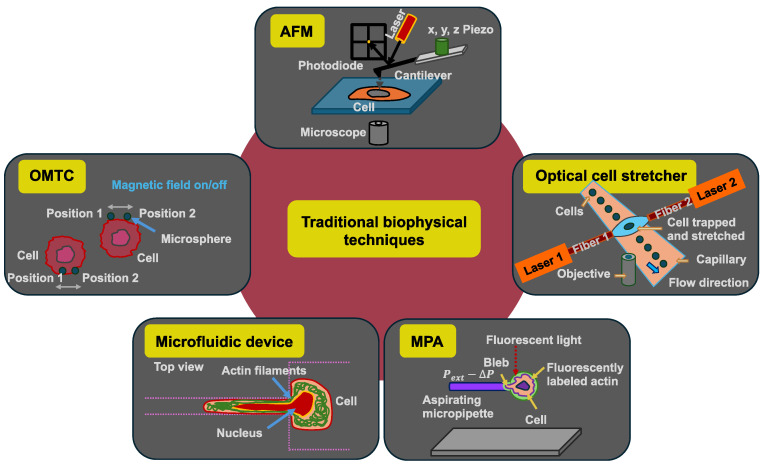
Schematic drawings of the five selected traditional biophysical techniques for the analysis of mechanical properties of individual cancer cells: atomic force microscopy (AFM), optical cell stretcher, micropipette aspiration (MPA), microfluidic device with constriction, and optical magnetic twisting cytometry (OMTC).

**Figure 4 cells-14-00584-f004:**
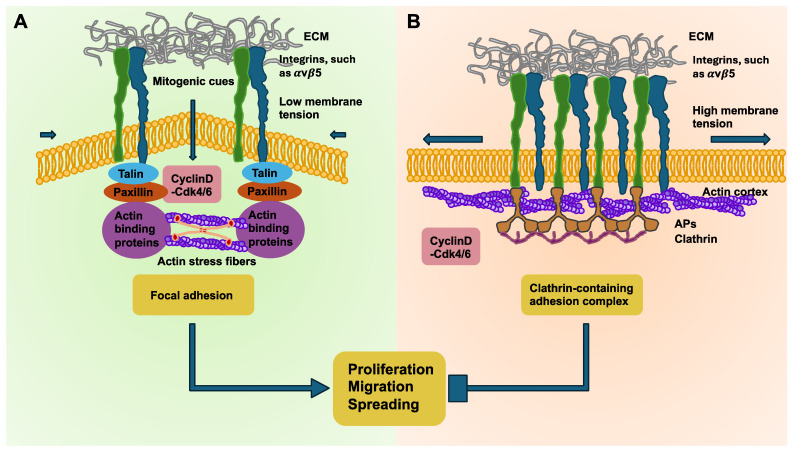
Tensional homeostasis of breast cancer cells. The adhesion mode controlled by membrane tension is crucial for the rate of proliferation, migration, and spreading. (**A**) The schematic sketch shows that high membrane tension together with cortically positioned actins and decreased cyclin D-Cdk4/6 activity causes coupling of the αVβ5 integrin adhesion sites with clathrin-containing adhesion complexes (CCACs). Therefore, the motility, spreading, and proliferation of the cells is impaired. (**B**) Upon decreasing membrane tension, actins self-organize into stress fibers, and when cyclin D-Cdk4/6 is active, such as in the early G1 phase of the cell cycle, CCACs uncouple, permitting the formation of focal adhesions at integrin αVβ5 sites, easing motility, spreading, and proliferation. Aps = adaptor proteins.

**Figure 5 cells-14-00584-f005:**
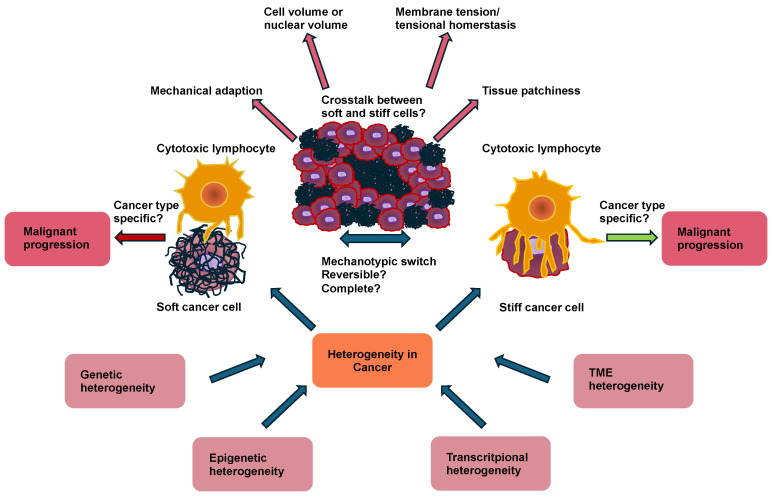
Mechanical markers of cancer cells are cancer type, cancer stage, and cancer localization specific. Since cancers are heterogeneous at various length scales, such as genetic, epigenetic, transcriptional, and TME heterogeneity, the degree of heterogeneity can influence the mechanotype of individual and cancer cells in the aggregate. The softness or stiffness of cancer cells can impact their interaction with cytotoxic lymphocytes, whereby the stiffer cells are more prone to elimination. Nevertheless, both stiff and soft cancer cells can contribute to the malignant progression of cancer and consequently cancer metastasis. Cancer cells seem to be able to switch between their mechanotype, which may be a complete switch that may be reversible. In the cancer tissue, cancer cells interact with one another such as soft and stiff cancer cells. The mechanical characterization of cancer cells, considering their environment, may include mechanical adaptation, changes in cell volume or nuclear volume, membrane tension or tension homeostasis, and tissue stiffness.

**Table 1 cells-14-00584-t001:** Softer cancer cells are more malignant compared with healthy counterparts. Cells were measured as individual cells.

Cancer Type	Cells	Technique + Ref.	Year
Bladder urothelial cancer	Normal Hu609 and HCV29 and bladder cancer cells (Hu456, T24 and BC3726 cells transformed by v-ras oncogene	AFM [4,12]	1999 and 2012
Bladder urothelial cancer	Normal human urothelial SV-HUC-1 and bladder cancer MGH-U1 cells	AFM [13]	2014
Bladder urothelial cancer	Normal SVHUC-1 control cells and TSGH8301 and J82 bladder cancer cells	AFM [14]	2015
Bladder urothelial cancer	Non-malignant bladder HCV29 cells and cancerous cells (HTB-9, HT1376, and T24)	AFM [15]	2014
Bladder urothelial cancer	Non-malignant cell cancer of the ureter (HCV29), bladder carcinoma (HT1376), and transitional cell carcinoma (T24)	AFM [16]	2022
Breast cancer	MCF-7 and benign MCF-10A	AFM [17]	2008
Breast cancer	Highly invasive MDA-MB-231, non-invasive MCF-7 and benign MCF-10A	AFM [18]	2015
Breast cancer	Highly invasive MDA-MB-231, non-invasive MCF-7 and benign MCF-10A	AFM [19]	2017
Breast cancer	MCF-7 and benign MCF-10A	Microfluidics [20]	2009
Breast cancer	Invasive transformed MCF-7 and non-metastatic MCF10A and non-transformed MCF-7	Optical cell stretcher [3]	2005
Breast cancer	MCF-10A (healthy), MCF-7 (tumorigenic/noninvasive), and MDA-MB-231 (tumorigenic/invasive)	AFM [21]	2016
Breast cancer	Primary suspended metastatic breast cancer cells	AFM [22]	2008
Breast cancer	Normal mammary tissue A184A1, T47D pleural effusion of an infiltrating ductal carcinoma of the breast, and MCF-7 breast adenocarcinoma	AFM [12]	2012
Breast cancer	Normal M10 control cells and MCF-7 and MD-MB-468 breast cancer cells	AFM [14]	2015
Breast cancer	Primary HMEpC cells from healthy epithelial breast tissue, fibroadenoma (FA, benign) and breast cancer cells	Optical cell stretcher [23]	2022
Cervical ovarian cancer	Highly invasive ovarian cancer cells (HEY A8), their less invasive parental cells (HEY), ovarian cancer OVCAR-3 and OVCAR-4 cells and immortalized ovarian surface control epithelial cells (IOSE)	AFM [24]	2012
Cervical ovarian cancer	Primary epithelial control cells and SiHa and HeLa cervical cancer cells	AFM [14]	2015
Cervical ovarian cancer	Primary normal cervical epithelial cells and cervical carcinoma cells	Optical cell stretcher [23]	2022
Colon cancer	HT-29 control and CX1 cancer cells	AFM [12]	2012
Esophagus cancer	Normal squamous cells (EPC2), metastatic (CP-A) and dysplastic (CP-D) esophageal cells	AFM [25]	2011
Ewing sarcoma	MSCs	AFM [26]	2023
Fibrosarcoma	HT-1080 and control fibroblasts	AFM [26]	2023
Kidney cancer	Carcinoma A-498 cells and adenocarcinoma ACHN cells and non-tumorigenic RC-124	AFM [27]	2013
Melanoma	WM793 control and 1205Lu melanoma cells; WM115 control and WM266-4 melanoma cells, WM35 control and A375 melanoma cells	AFM [12]	2012
Osteosarcoma	SaOs-2 and control human primary bone marrow-mesenchymal stem cells (MSCs) were isolated, differentiated and propagated into osteoblasts	AFM [26]	2023
Pancreas cancer	HPDE control cells and BxPC-3, PANC-1, ASPC-1, and MiaPaca-2 pancreas cancer cells	AFM [14]	2015
Prostate cancer	Non-tumorigenic prostate cells PZHPV-7, prostatic adenocarcinoma initiated from bone metastasis PC-3, metastatic prostate carcinoma from brain metastasis Du145, metastatic prostate carcinoma established from the left supraclavicular lymph node metastasis LNCaP	AFM [12]	2012
Prostate cancer	Primary benign prostate hyperplasia (BPH) and two prostate cancer cell lines LNCaP clone FGC and PC-3	AFM [28]	2008
Rhabdomyosarcoma	RD and control skeletal muscle cells (SKMCs)	AFM [26]	2023
Thyroid cancer	Primary thyroid S748 cells, anaplastic carcinoma S277 cells and normal control cells	AFM [29]	2012

**Table 2 cells-14-00584-t002:** Stiffer cancer cells are more malignant compared with healthy counterparts. The cells were analyzed as individual cells.

Cancer type	Cells	Technique + Ref.	Year
Cervical squamous carcinoma	Exfoliated cells were collected from nine patients with chronic cervicitis or cervical intraepithelial neoplasia 1 (CIN1) (control group), 30 patients with CIN2–3 (CIN 2–3 group), and 13 patients with cervical cancer (cervical cancer group)	AFM [99]	2015
Chondrosarcoma	SW1353 and control primary chondrocytes from femoral condyles of healthy controls	AFM [26]	2023
Hepatocellular carcinoma	Hepatocellular carcinoma (HCC) cells and normal hepatocytes	Micropipette aspiration technique [100]	2000
Hepatocellular carcinoma	Hepatocellular carcinoma (HCC) cells and normal hepatocytes	Micropipette aspiration technique [41]	2002
Leukemia	Lymphocytes from chronic lymphocytic leukemia (CLL) patients and healthy donor	Microfluidic device and AFM [101]	2015
Melanoma cells	Highly motile non-metastatic B16-F10 cells have low cell stiffness, and low motile and metastatic B16-F1 cells have high cell stiffness	AFM [102]	2012
Myeloma	Myeloid (HL60) cells were measured to be a factor of 18 times stiffer than lymphoid (Jurkat) cells and six times stiffer than human neutrophils on average	AFM [103]	2006
Prostate cancer cells	Progressively increasing metastatic potential cell lines (22RV1, LNCaP, DU145, and PC-3),	Optical magnetic twisting cytometry (OMTC) [104]	2022
Prostate cancer cells	Lowly (LNCaP) and highly (CL-1, CL-2) metastatic human prostate cancer cells	AFM [105]	2012
Prostate cancer cells	Lowly LNCaP, intermediate DU145, and highly metastatic PC3 cells	Microfluidic device [106]	2019

**Table 3 cells-14-00584-t003:** Major strength and weaknesses/limitations of various techniques.

Technique	Strength	Weaknesses/Limitations
**AFM**	Measures cells in an adherent state [117]; Delivers high-resolution nanotopographic images [118]; Needs only relatively uncomplicated sample processing; Repeated measurements of the same cell can be performed at the same or different positions; The impact of cell adhesion can be addressed; Capable of both nanometer-resolution scanning and micrometer-range bulk measurements; Can measure cells with indentations ranging from hundreds of piconewtons to several micronewtons of force.	Cells can be measured at a specific location (nuclear vs. cytoskeletal region), which may lead to different stiffness measurements [124]; Requires controlled vibrational environment; Needs spring constant calibration for each cantilever [125]; Procedure with low throughput that demands substantial technical expertise from users; The technology is highly complex and requires high-quality equipment.
**Optical cell stretcher**	Measures cells in a non-adherent state [3,30]; Cell contact-free force is exerted on the cell’s surface [3,30]; Different measurement protocols are possible; Mechanical properties of the cell in its non-adherent state can be determined.	Temperature can alter the measurement [126]; Measurement of granulated cells is not possible such as melanoma cells; The impact of cell adhesion cannot be addressed; A cell can only be measured once; The force generated through optics is low, therefore limiting the cell types that can be mechanically measured.
**MPA**	Cells can be measured in an adherent and non-adherent state; Easy and straightforward to use.	It is restricted in its usage due to the low spatial and temporal resolution and the consequent large deformations imposed on the cell [127]; Operating experience is relied upon, and precision is not measured adequately [128]; It faces the challenges of bulky equipment and low throughput [129].
**Microfluidic devices**	Automation reduces user subjectivity, improves reproducibility and reduces labor-intensive tasks [130]; Measures cells in a non-adherent state; A wide range of microfluidic system designs are possible, allowing the cells to be mechanically analyzed multiple times; There is the possibility of obtaining large amounts of data; In tests based on constriction, the impact of cell size on the transition time was factored in [92].	The measured variables are subject to the adhesion behavior of the cells and the multi-parameter measurement cannot be evaluated qualitatively with other techniques [130]; A non-contact measurement method; A potentially destructive examination and possible obstruction of the passageway.
**OMTC**	Permits the analysis of cell characteristics on micrometer–nanometer length scales; Multiple cells can be analyzed in parallel; Probes a specific type of receptor through specific coating of the magnetic microspheres.	A positional deviation of the magnetic beads can lead to substantial measurement errors and limited measurement precision [131]; Differences in the expression of the investigated receptors on the cell surface can influence the result.

## Data Availability

Not applicable.

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
