# Peer review of "Softness or Stiffness What Contributes to Cancer and Cancer Metastasis?"

_cells, 2025, doi:10.3390/cells14080584_

Round 1

Reviewer 1 Report

Comments and Suggestions for Authors

The review highlights the main advances in the mechanical characterization of cancer cells, discusses the strengths and weaknesses of the current approaches, and questions whether passive mechanical characterization of cancer cells is still considered state-of-the-art. The text is well written and can be accepted after a few minor corrections.

On page 195, the following phrase is repeated and should be corrected:
"Cancer cell membranes usually have higher cholesterol levels compared to normal cells, which renders cancer cells soft. Subsequently, a mechanical immune checkpoint inhibitor (MICI) was developed, which..."

In Table 3, a reference should be included in the “Micropipette aspiration” section under weaknesses/limitations.

In Table 3, within the microfluidics section, there is a mistake where a phrase indicating “hours” was inadvertently added.

Additionally, what is the meaning of “RGD” in section 3.5? Please clarify.

Finally, please review the list of abbreviations, as linear dichroism was not mentioned in the text.

Author Response

Dear Reviewer 1

Thank you very much for your extremely helpful comments on my review article. I have taken all your comments and suggestions into account. I have highlighted the changes in yellow and also provide a clear version of my manuscript.

My answers to Reviewer 1’s comments:

Reviewer 1:

The review highlights the main advances in the mechanical characterization of cancer cells, discusses the strengths and weaknesses of the current approaches, and questions whether passive mechanical characterization of cancer cells is still considered state-of-the-art. The text is well written and can be accepted after a few minor corrections.

Comment 1. On page 195, the following phrase is repeated and should be corrected:

"Cancer cell membranes usually have higher cholesterol levels compared to normal cells, which renders cancer cells soft. Subsequently, a mechanical immune checkpoint inhibitor (MICI) was developed, which..."

Answer 1: Thank you for pointing this out. I have reworded it and deleted the repetition.

Comment 2. In Table 3, a reference should be included in the “Micropipette aspiration” section under weaknesses/limitations.

Answer 2: Thank you. I have included the citation.

Comment 3. In Table 3, within the microfluidics section, there is a mistake where a phrase indicating “hours” was inadvertently added.

Answer 3: Thank you. I have deleted it.

Comment 4. Additionally, what is the meaning of “RGD” in section 3.5? Please clarify.

Answer 4: Thank you. It is the RGD peptide, in which the three amino acids arginine (R), glycine (G), and aspartic acid (D) are linearly arranged.

Comment 5. Finally, please review the list of abbreviations, as linear dichroism was not mentioned in the text.

Answer 5: Thank you. I have reviewed and corrected the list of abbreviations.

Best regards

Claudia Tanja Mierke

Reviewer 2 Report

Comments and Suggestions for Authors

The contribution of softness to cancer progression and metastasis has been a constant debate and not resolved issue. The authors have put together an excellent review charting various findings in this area to resolve the dilemma. Importantly, various techniques helpful to resolve this are also described in this article. I recommend the article to be published.

Author Response

Dear Reviewer 2

Thank you very much for your extremely positive review of my review article. Reviewers 1 and 3 had minor requests for changes, which I fully complied with. I have highlighted the changes in yellow and also provide a clear version of my manuscript. 

My answer to Reviewer 2’s comments/review:

Reviewer 2: The contribution of softness to cancer progression and metastasis has been a constant debate and not resolved issue. The authors have put together an excellent review charting various findings in this area to resolve the dilemma. Importantly, various techniques helpful to resolve this are also described in this article. I recommend the article to be published.

Answer: Thank you for the extremely positive review.

Best regards

Claudia Tanja Mierke

Reviewer 3 Report

Comments and Suggestions for Authors

This is an excellent review that addresses an aspect of basic oncology that is still in its infancy, regarding the importance of cell softness/stiffness in tumor progression. The author presents the state of the art in the methodologies available for determining cell stiffness, as well as the advantages and disadvantages of each method. The contradictory role of cell stiffness, which is sometimes protumor and sometimes antitumor, is very well discussed. The discussion on the patchiness model is really outstanding.

I noticed only a few typos, which I list below.

Table 3: A citation is missing in the last line of page 12.

Table 3: page 13, 2nd column. Please delete date and hour

Page 15, line 453. Please fix the word microfluidic

Page 16, line 520, fix the word trypsination

Page 18, line 601. Fix tensional homeostasis

Author Response

Dear Reviewer 3

Thank you very much for your extremely helpful comments on my review article. I have taken all your comments and suggestions into account. I have highlighted the changes in yellow and also provide a clear version of my manuscript.

My answers to Reviewer 3’s comments:

Reviewer 3: This is an excellent review that addresses an aspect of basic oncology that is still in its infancy, regarding the importance of cell softness/stiffness in tumor progression. The author presents the state of the art in the methodologies available for determining cell stiffness, as well as the advantages and disadvantages of each method. The contradictory role of cell stiffness, which is sometimes protumor and sometimes antitumor, is very well discussed. The discussion on the patchiness model is really outstanding.

I noticed only a few typos, which I list below.

Comment 1. Table 3: A citation is missing in the last line of page 12.

Answer 1: Thank you. The citation is now included.

Comment 2. Table 3: page 13, 2nd column. Please delete date and hour

Answer 2: Thank you. I have deleted it.

Comment 3. Page 15, line 453. Please fix the word microfluidic

Answer 3: Thank you. I have made the correction.

Comment 4. Page 16, line 520, fix the word trypsination 

Answer 4: Thank you. I have corrected it.

Comment 5. Page 18, line 601. Fix tensional homeostasis

Answer 5: Thank you. I have corrected it.

Best regards

Claudia Tanja Mierke
